# Optimal Pressure Management in Water Distribution Systems: Efficiency Indexes for Volumetric Cost Performance, Consumption and Linear Leakage Measurements

**Risimati Patrick Mathye [1,\*]** , **Miklas Scholz [1,2,3,4]** and **Stephen Nyende-Byakika [5]**

1. Department of Civil Engineering Science, University of Johannesburg, Auckland Park, P.O. Box 524, Johannesburg 2006, South Africa; miklas.scholz@tvrl.lth.se
2. Division of Water Resources Engineering, Faculty of Engineering, Lund University, P.O. Box 118, 221 00 Lund, Sweden
3. Department of Town Planning, Engineering Networks and Systems, South Ural State, 76, Lenin Prospekt, Chelyabinsk 454080, Russia
4. Institute of Environmental Engineering, Wrocław University of Environmental and Life Sciences, 25 Norwida St., 50-375 Wrocław, Poland
5. Department of Civil Engineering, Water Sisulu University, Old Berlin Road, Fort Jackson, P.O. Box 1421, East London 5200, South Africa; stenbyak@gmail.com
\* Correspondence: risimatimathye@gmail.com

**Abstract:** Water is a finite resource and should be given the attention it deserves to reduce its depletion through leakages in pipe systems. The authors implemented pressure management strategies linked to fixed and variable discharge (FAVAD), the burst and background estimate (BABE), and orifice principle methodologies to analyze a two-phased comparative method for applying optimal pressure management and its efficiency indexes in measuring volumetric cost performance, consumption, leakage flowrate, linear leakage reduction, infrastructural leakage and leakage cost indices. Using time-modulated smart control pressure reducing valve (PRV) simulation processes, the authors selected Alexandra Township in Johannesburg, South Africa as a case study. The results showed a reduction in head pressure, a reduction in the system input volume (SIV) from 26,272,579 m$^3$ to 21,915,943 m$^3$ and a reduction in minimum night flow (MNF) from 14.01% to 12.50%. The annual estimated nodal system output (NSO) was reduced from 14,774.62 m$^3$ to 12,787.85 m$^3$. The monthly average linear system repairs were reduced from 246 to 177, while the efficiency index percentages of leakage frequency/km/pressure were reduced from 8.31% to 5.98%. At a unit cost of $3.18/m$^3$, the cost of leakages declined from $4,009,315.54 to $2,862,053.10 per month, while average household consumption (AMC) reduced from 36.33 m$^3$ to 24.56 m$^3$. Finally, the linear reduction value R$^2$ for the percentage of the total leakage flowrate (TLFR)/SIV declined from 0.58 to 0.5, whereas the infrastructure leakage ratio (ILI) increased from 4 to 4.3. The results fully demonstrated that optimal pressure management is an alternative way to simulate, estimate, quantify and understand where and how water is lost in a distribution system. The authors propose that the implementation of proactive leakage management and domestic background leakage repair could further assist in reducing the frequency and cost of water leakages.

**Keywords:** optimal pressure management; linear leakage measurements; volumetric efficiency indexes; customer consumption saving; water cost reduction

## 1. Introduction

Water is a vital and fundamental resource for human health, survival and development. However, the world's water resources continue to be depleted by high customer demand and infrastructural losses and leakages [1–3]. Although water leakages continue to increase in developed and developing countries, detecting and repairing them is even more costly for most developing countries experiencing limited financial capacity [2–4]. As a result of

this realization, there has been a significant global change among water managers who are determined to reduce water leakages. In practice, one of the trusted methods to reduce water leakages is pressure management control [5–7]. Pressure management is a basic tool to manage water leakages in a water distribution system because the leakage flowrate is a function of change in head pressure or average zonal pressure [8–12]. In leakage hydraulics, many researchers are of the view that a change in average zonal pressure (AZP) is a direct integration of the fixed and variable discharge principle (FAVAD), the burst and background estimate (BABE) and the orifice principle, which are all efficient methodologies for assessing water leakage behavior [2,6,12–16]. In relation to pressure management, some of the most used tools are pressure reduction methods which use a time-modulated or flow-modulated PRV in a selected network zone or nodal output to control upstream and downstream pressures [8,9,17,18]. Among the noted benefits of pressure management in leakage reduction are that it

- reduces working pressure, which helps to conserve water;
- improves the reliability of the continued supply by reducing pipe bursts;
- reduces the fluctuation of pressure in the system;
- increases the lifespan of the water supply assets;
- decreases the costs of operations through a reduction in burst frequency as well as energy consumption;
- is efficient with respect to water demand and conservation management; and
- offers possible cost savings through pressure reduction options [14,19–21].

Pressure management has been used with great success in some parts of the world [6,8,10,12,22,23]. However, other studies indicate that pressure management approaches may be difficult to implement in developing countries characterized by theft, illegal connections, old infrastructures, poverty and unplanned settlements due to socio-economic factors [2,4,9,16,24–26]. There are, however, multiple studies in some parts of the world that looked at the application of PRV as an energy recovery solution for leakage control [16,27]. Some studies also proved that a reduction of the operational costs in leakage control is directly proportional to the application of PRV in leakage control [9,14,18,24]. The use of pressure management as a predictive solution to assess pipe failure in water distribution systems has also been implemented in some parts of the world. Findings indicate that pressure management helps to increase the useful investment life of infrastructures [28,29].

With reference to multiple studies in water leakage control through the use of pressure management approaches, the objective of this paper is to show the influence of optimal pressure management in water distribution systems through efficiency indexes. The measurement indexes were applied through a comparative analysis of two phases (before and after the application of optimal pressure management) and are as follows: leakage/km/AZP, volumetric cost efficiency performance, customer consumption, SIV, MNF, total leakage flowrate (TLFR), linear leakage repair reduction, the economics of leakages and ILI. For illustration purposes, Alexandra Township, which is located in the north-eastern part of Johannesburg, South Africa and has the coordinates 26°6′1.68″ S and 28°7′3.50″ E, was selected [30]. Alexandra Township is a socio-economically deprived township with a high unemployment rate [20–32]. For example, in the year 2016, household consumption was 52% higher than a set limit of 20 m$^3$/month [30]. The report further noted a high percentage of water losses due to illegal connections, which is attributed to existing socio-economic problems, such as high unemployment and poverty. The area's non-revenue water was recorded at 87.02% of the total system's input volume, equating to a total cost of $49.882 million USD dollars, whereas MNF was estimated at over 70% [33] The authors used Alexandra Township as a suitable area to demonstrate the significance of optimal pressure management in water distribution systems by measuring its efficiency indices in terms of volumetric cost performance, customer consumption and linear leakage measurements.

## 2. Methodology and Materials

The proceeding sections present the methodology followed by the researchers in this study—a combination of a literature search, system data analysis and visual condition assessment on site. The general methodolgy applied in this study is based on the variables fixed and variable discharge (FAVAD), burst and background estimates, unavoidable annual real losses (UARL), current annual real losses (CARL), infrastructure leakage index (ILI), burst and background estimates (BABE) as well as minimum night flow (MNF) [8,24,26]. The average zonal pressure (AZP) and flows at each node were monitored using the Supervisory Control and Data Acquisition (SCADA), Water Distribution and System Optimization (WADISO) and Infrastructure Monitoring Query System (IMSQ) methods. The existing hydraulic model was verified for each district metered area (DMA) using the data provided by the water utility's (Johannesburg Water SOC LTD) SCADA and onsite data verifications by the authors. The results were compared with the findings of this study as presented in the proceeding sections.

### 2.1. Preliminary Data Collection and Hydraulic Simulation Process

The pre-data flow logging approach used in this study involved:

- a literature search to collect historic loss levels;
- the use of SAP-PM, a customer-centric software application for tracking infrastructure leakage failures, for periods between 2015 and 2019; and
- the use of ultra-sonic flow and pressure logging devices to measure preliminary flow and operating system pressures from the six supplying DMA connections and their flow-modulated PRV.

All flows and pressures were recorded for 15 days and used to compute the total system input volume (SIV) and minimum night flow (MNF) between 12:00 a.m. and 4:00 a.m. [28,34]. The authors further checked flow and pressure from 20 critical nodal points (CNPs) in the distribution system within the six DMAs and finally used the Water Distribution and System Optimization's (WADISO)_(GLS Software (PTY) Ltd., South Africa) hydraulic modelling software application (a similar group software product to EPANET 2.2 (US EPA Research, Durham, NC, USA)) and Infrastructure Monitoring Query System (IMSQ) software to compare onsite preliminary collected flow and pressure values from six DMAs and 20 CNPs against the approved and designed hydraulic model as per WADISO. Finally, the researchers conducted customer water consumption estimations per household through daily meter readings of metering devices for a period of 7 days.

The preliminary findings outlined above were used to develop the methodological flow process as displayed in Figures 1 and 2. In order to achieve the study objectives, all the changes in average flow, operating pressure, customer consumption patterns and infrastructure leakage trends, as well as the reduction in volumetric SIV and MNF rates, were measured, simulated and analyzed in two phases, as presented in the proceeding sections. The authors used WADISO, IMQS (IMQS SOFTWARE (PTY) LTD, Johannesburg, South Africa) and Excel spreadsheets to tabulate, analyze, simulate and create a graphical presentation of all the collected data.

### 2.2. Logging and Simulation of the Transient Data Flow and the Indexes' Computation

Phases 1 to 3, described below, present the practical methodological approaches followed in this study. The process flow presented in Figure 1, below, was executed twice after preliminary data collection for purposes of establishing the effects of pre- and post-optimal pressure adjustment in leakage control measurements. The data recording, simulation and analysis processes in this study were conducted using ultrasonic flow and pressure data loggers.

2.2.1. Phase 1: Flow and Pressure Simulation Process

Phase 1 included steps (1) to (3) as per Figure 1. This entailed setting up the hydraulic bulk flow data logging process for simulation of transient flows from the six DMAs and

20 CNPs for a period of 15 days between 21 September 2020 and 6 October 2020. Flow and pressure were recorded every 30 min and every four hours for each DMA and CNP, respectively. We then applied the orifice principle, FAVAD, BABE and MNF methodologies to analyze changes in average pressure, average flowrate, leakage trends per kilometer of pipeline as well as average household customer consumption. This exercise was aimed at establishing any changes in average flowrate with respect to system input volume (SIV), MNF and nodal system output (NSO) flows from the DMA and CNP in the distribution system. Pre-logging led to the three flow-modulated PRVs marked LP-1, LP-2 and LP-3 (Figure 2) exhibiting higher average zonal pressures (AZP) of between 9 and 18 bar when verified through the SCADA's WADISO and IMQS systems. The following section presents a further outlook on the three dysfunctional PRVs.

2.2.2. Phase 2: Simulation Process

After completion of Phase 1, the research team implemented steps 3 and 4 as per Figure 1. All hydraulic models in step 3 were used to interpret, simulate and analyze all transient flows abstracted from the three flow-modulated PRVs marked as LP-1, LP-2 and LP-3 (Figure 2). We then recommended that the three critically identified flow-modulated PRVs should be replaced by time-modulated smart control PRVs. The time-modulated smart control PRVs were calibrated for pressure and flows measurements and re-aligned to the stipulated downstream pressures of between 2 and 9 bar as per WADISO. The time-modulated smart control PRVs were chosen because they allowed for the pre-setting of the PRVs for minimum automated time-modulated flows through reduced pressure during off-peak periods measured or MNF durations between 12:00 a.m. and 4:00 a.m.

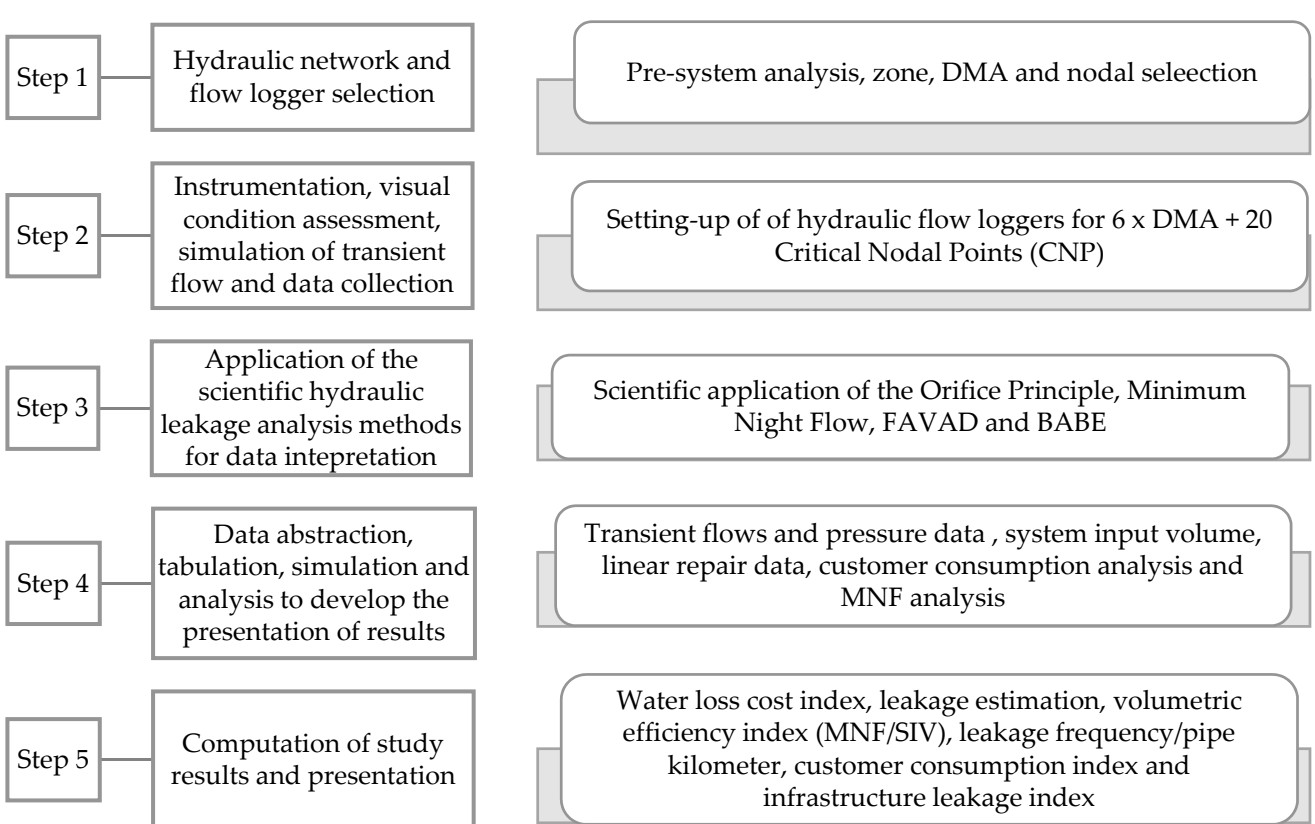

**Figure 1.** Process flow and analysis methods for the case study.

In phase 2, the authors introduced six boundary zone valves (closed zone valves) to separate the two bulk supply zones and the six DMAs. The time-modulated smart control PRV downstream pressures were later adjusted to 2 and 9 bar as per the WADISO hydraulic model system. Similar to phase 1, the team then re-logged all flows using ultra-sonic flow

and pressure logging devices and simulated pressure and flow data for a period of 15 days between 22 February 2021 and 9 March 2021 at all DMAs and CNPs.

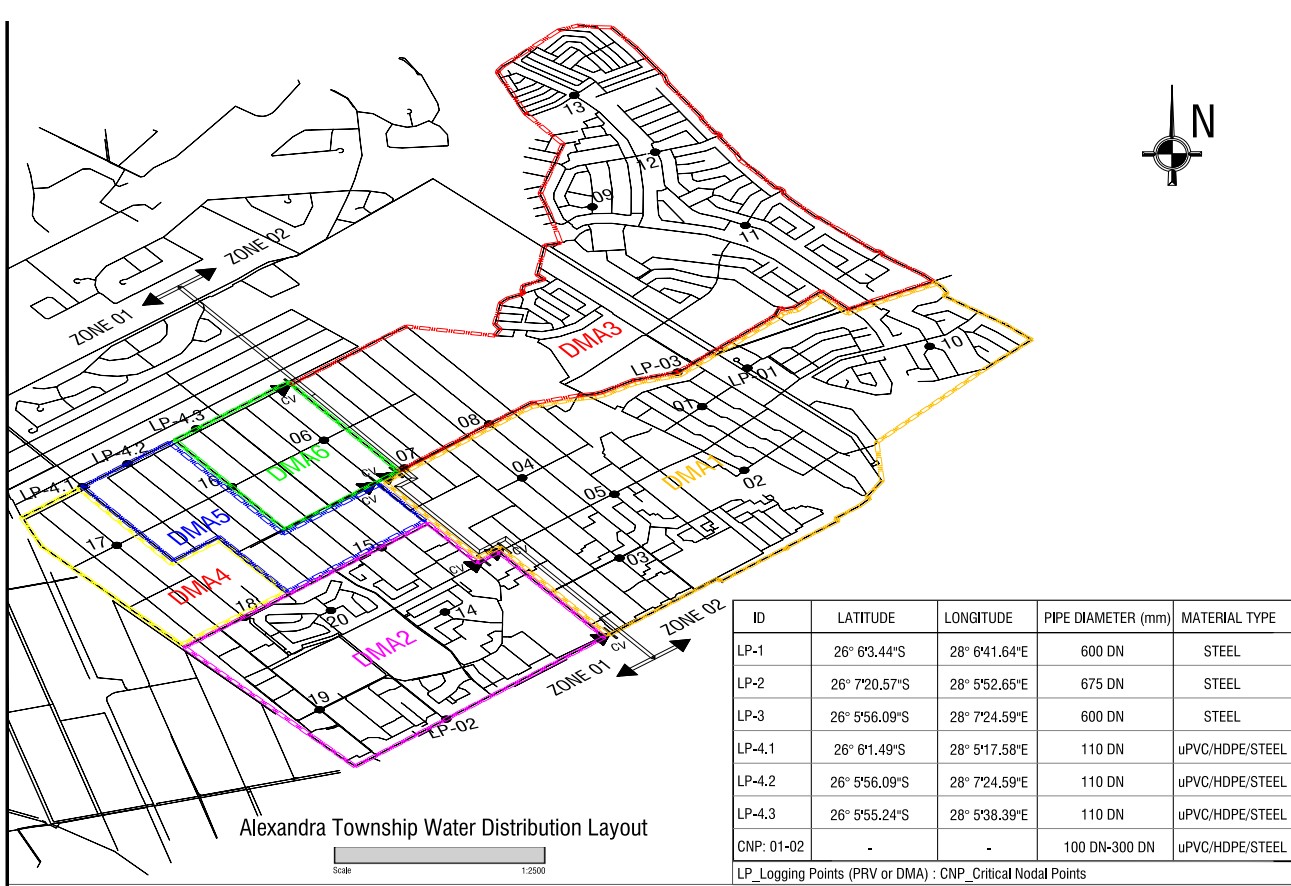

| ID | LATITUDE | LONGITUDE | PIPE DIAMETER (mm) | MATERIAL TYPE |
|---|---|---|---|---|
| LP-1 | 26° 6'3.44"S | 28° 6'41.64"E | 600 DN | STEEL |
| LP-2 | 26° 7'20.57"S | 28° 5'52.65"E | 675 DN | STEEL |
| LP-3 | 26° 5'56.09"S | 28° 7'24.59"E | 600 DN | STEEL |
| LP-4.1 | 26° 6'1.49"S | 28° 5'17.58"E | 110 DN | uPVC/HDPE/STEEL |
| LP-4.2 | 26° 5'56.09"S | 28° 7'24.59"E | 110 DN | uPVC/HDPE/STEEL |
| LP-4.3 | 26° 5'55.24"S | 28° 5'38.39"E | 110 DN | uPVC/HDPE/STEEL |
| CNP: 01-02 | - | - | 100 DN-300 DN | uPVC/HDPE/STEEL |

LP_Logging Points (PRV or DMA) : CNP_Critical Nodal Points

**Figure 2.** The district's metered areas as well as flow and pressure logging simulation points.

### 2.2.3. Phase 3: Simulation Process and Computation of Efficiency Indexes

All collected data and hydraulic simulations after the change in optimal pressures from a range of 9 to 18 bar to a lower range of 2 to 9 bar were used to compare results between phase 1 and phase 2. As presented in Figure 2, the following results were derived accordingly:

- leakage flowrate ratio;
- leakage frequency/km/pressure linear repair data;
- change in volumetric flow;
- the ratio of MNF/SIV;
- changes in customer consumption; and
- water-saving costs in comparison to the findings of Phase 1.

The mathematical formulations for the efficiency indexes are presented in the proceeding Section 2.3.

Figure 1 shows the methodological flow process that the authors followed in this study and Figure 2 indicates the data flow and pressure logging simulation points for each DMA. The simulated data were based on the orifice principle and thereby used to estimate the SIV, MNF and BABE parameters. The team incorporated the FAVAD principle to compute the total leakage flowrates and volumetric leakage indexes for the study. The changes in optimal pressure at the three time-modulated smart control PRVs were used as a base to compute and compare the pre- and post-efficiency index ratios, as presented in stages 4 and 5 of Figure 1.

Figure 2 shows the hydraulic flow and pressure simulation layout plan that was set up for the study area. The figure shows zonal supply areas (Zone 01 and Zone 02), six logging points (LPs), marked LP-01, LP-02, LP-03, LP-4.1, LP-4.2 and LP-4.3, and their subsequent six DMAs. There are also 20 CNPs within the six DMAs (marked 01 to 20), which were pre- and post-logged to measure and simulate changes in flow and average zonal pressure (AZP) after the optimal pressure adjustment through a time-modulated smart control PRV.

Figure 3 and Table 1 shows the visual condition assessment flow chart process followed in this study and the water distribution characteristics. The visual condition assessment is a base which the authors used to verify the water distribution system in order to practically confirm the site conditions of the study area and compare the findings with data from IMQS and WADISO. The onsite visual condition assessment of PRVs and CNPs was conducted with the water utility's maintenance crews and involved, amongst other procedures, practical testing of zone valves, inspection of flow modulated PRVs and recording upstream and downstream pressures for all DMAs and the 20 CNPs within the distribution system.

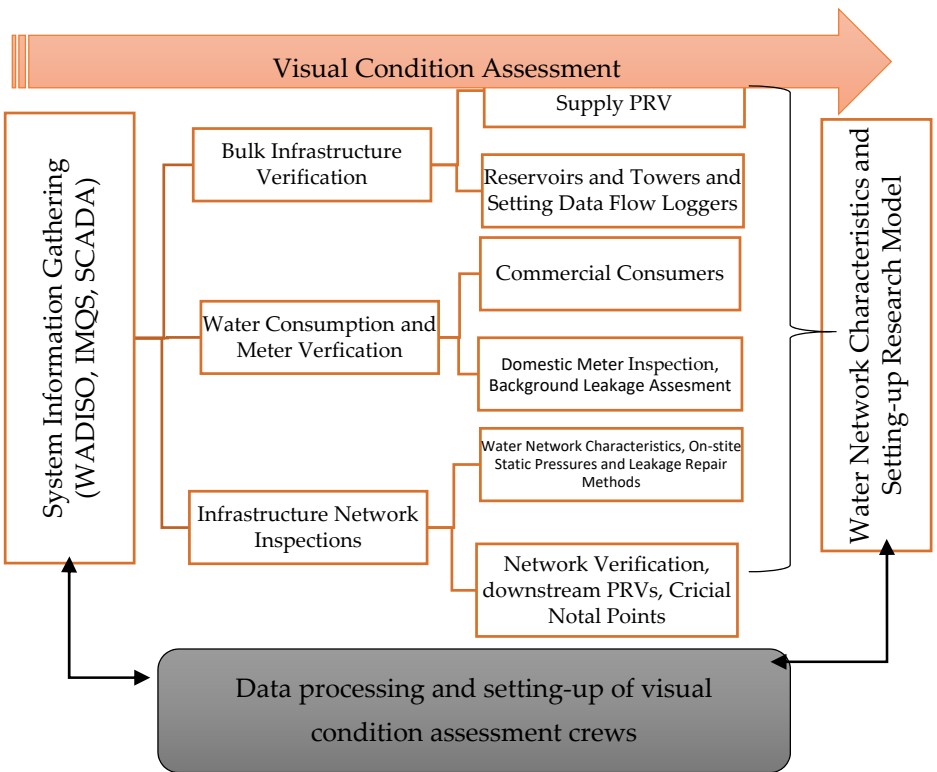

**Figure 3.** Visual condition assessment process for water leakage measurements.

Table 1a–c present the preliminary data collection conducted for the development of the water distribution network characteristics. The data show that three reservoirs supply the area and that the majority of the pipelines are made of uPVC and steel (older than 25 years). The varying pipeline materials have different coefficients of expansion due to pressure changes. The average dynamic pressure in the area was 180 m, while most pipes are pressure-rated class 16 and 9. It was also confirmed that the topographical layout of the study area is what makes the energy grade line (EGL) steep and hence reduces high pressures when measured from the reservoir's top water level (TWL). The preliminary data findings were further confirmed by a high number of pipeline leakage observations during visual condition assessments. The authors used this visual condition assessment to draw up the research methodology and mathematical formulations for the study.

**Table 1.** (a) Water reservoir characteristics, (b) water distribution characteristics (DMA), and (**c**) critical nodal point (CNP) characteristics analysis.

**(a)**

| Name of Reservoir | Elevation | Top Water Level (m) | Static Head (m) | Latitude | Longitude | DMA Supply | Node Supply |
|---|---|---|---|---|---|---|---|
| Linbro Park | 1617.47 | 1642.47 | 100.08 | 26°10′2.90″ S | 28°13′2.89″ S | DMA1, DMA3 | 1, 2, 3, 4, 5, 8, 9, 10, 11, 12, 13 |
| Marlboro | 1592.6 | 1600.12 | 60.00 | 26°09′4.38″ S | 28°08′7.11″ S | DMA4, DMA5, DMA6 | 6, 7, 15, 16, 17, 18 |
| Randjieslaagte | 1667.64 | 1674.6 | 70.03 | 26°14′1.65″S | 28°09′1.55″S | DMA2 | 14, 19, 20 |

**(b)**

| | | Pipeline Details | | | | | | | Average Pressure Outlook | | |
|---|---|---|---|---|---|---|---|---|---|---|---|
| ID | PRV Size | Pipe Diameter (mm) | Pipe Material | Pipe Age (Years) | Number of Nodes (DMA) | Energy Grade Line (m) | Elevation (m) | Co-Efficient of Expansion (K-2) | Head (TWL–PRV Elevation) (m) | Total Head (Static + Head Diff) | Dynamic Head (Total Head, EGL–TWL) (m) |
| LP-1 | 300 | 600 | Steel | 22 | +500 | 1596.38 | 1514.74 | $1.2 \times 10^{-5}$ | 127.73 | 227.81 | 181.72 |
| LP-2 | 200 | 675 | Steel | 59 | +500 | 1654.79 | 1599.34 | $1.2 \times 10^{-5}$ | 75.26 | 145.29 | 125.48 |
| LP-3 | 300 | 600 | Steel | 22 | +500 | 1595.93 | 1523.16 | $1.2 \times 10^{-5}$ | 119.31 | 219.39 | 172.85 |
| LP-4.1 | - | 110 | uPVC | 29 | −100 | 1610.26 | 1552.84 | $8 \times 10^{-5}$ | 47.28 | 107.28 | 117.42 |
| LP-4.2 | - | 110 | uPVC | 29 | −100 | 1610.27 | 1549.59 | $8 \times 10^{-5}$ | 50.53 | 110.53 | 120.68 |
| LP-4.3 | - | 110 | uPVC | 29 | 30–50 | 1610.3 | 1547.82 | $8 \times 10^{-5}$ | 52.30 | 112.30 | 122.48 |

**(c)**

| ID | PRV Size | Pipe Diameter (mm) | Pipe Material | Pipe Age (Yrs) | Number of Connection (Node) | Energy Grade Line (m) | Elevation (m) | Co-Efficient of Expansion (K-2) | Head (TWL–PRV Elevation) (m) | Total Head (Static + Head Diff) | Dynamic Head (Total Head, EGL–TWL) (m) |
|---|---|---|---|---|---|---|---|---|---|---|---|
| 1 | DMA1 | 110 | uPVC | 29 | 4 | 1595.94 | 1515.4 | $8 \times 10^{-5}$ | 127.07 | 227.15 | 180.62 |
| 2 | DMA1 | 110 | uPVC | 25 | 3 | 1595.63 | 1554.82 | $8 \times 10^{-5}$ | 87.65 | 187.73 | 140.89 |
| 3 | DMA1 | 110 | uPVC | 30 | 4 | 1611.33 | 1547.49 | $8 \times 10^{-5}$ | 94.98 | 195.06 | 163.92 |
| 4 | DMA1 | 110 | uPVC | 30 | 4 | 1611.28 | 1544.05 | $8 \times 10^{-5}$ | 98.42 | 198.5 | 167.31 |
| 5 | DMA1 | 110 | uPVC | 30 | 4 | 1613.39 | 1550.26 | $8 \times 10^{-5}$ | 92.21 | 192.29 | 163.21 |
| 6 | DMA6 | 160 | uPVC | 29 | 4 | 1627.38 | 1594.34 | $8 \times 10^{-5}$ | 5.78 | 65.78 | 93.04 |
| 7 | DMA6 | 160 | uPVC | 29 | 4 | 1627.38 | 1520.19 | $8 \times 10^{-5}$ | 79.93 | 139.93 | 167.19 |
| 8 | DMA3 | 160 | uPVC | 29 | 3 | 1595.62 | 1520.56 | $8 \times 10^{-5}$ | 121.91 | 221.99 | 175.14 |
| 9 | DMA3 | 200 | uPVC | 22 | 3 | 1577.84 | 1541.47 | $8 \times 10^{-5}$ | 101 | 201.08 | 136.45 |
| 10 | DMA1 | 100 | steel | 35 | 3 | 1595.27 | 1527.43 | $1.2 \times 10^{-5}$ | 115.04 | 215.12 | 167.92 |
| 11 | DMA1 | 110 | uPVC | 23 | 4 | 1570.34 | 1538.59 | $8 \times 10^{-5}$ | 103.88 | 203.96 | 131.83 |
| 12 | DMA1 | 110 | uPVC | 25 | 4 | 1568.2 | 1526.95 | $8 \times 10^{-5}$ | 115.52 | 215.6 | 141.33 |
| 13 | DMA1 | 110 | HDPE | 10 | 4 | 1562.66 | 1512 | $20 \times 10^{-5}$ | 130.47 | 230.55 | 150.74 |
| 14 | DMA2 | 100 | steel | 30 | 4 | 1614.75 | 1610.25 | $1.2 \times 10^{-5}$ | 64.35 | 134.38 | 74.53 |
| 15 | DMA5 | 300 | uPVC | 30 | 3 | 1595.81 | 1530.99 | $8 \times 10^{-5}$ | 111.48 | 211.56 | 164.9 |
| 16 | DMA5 | 110 | uPVC | 29 | 3 | 1610.43 | 1547.62 | $8 \times 10^{-5}$ | 94.85 | 194.93 | 162.89 |
| 17 | DMA4 | 110 | uPVC | 29 | 4 | 1659.84 | 1586.96 | $8 \times 10^{-5}$ | 55.51 | 155.59 | 172.96 |
| 18 | DMA4 | 100 | HDPE | 18 | 3 | 1610.86 | 1555.88 | $20 \times 10^{-5}$ | 44.24 | 144.32 | 155.06 |
| 19 | DMA2 | 160 | uPVC | 30 | 4 | 1661.51 | 1600.43 | $8 \times 10^{-5}$ | 74.17 | 144.2 | 131.11 |
| 20 | DMA2 | 100 | steel | 30 | 4 | 1613.75 | 1562.5 | $1.2 \times 10^{-5}$ | 112.1 | 182.13 | 121.28 |

## 2.3. Mathematical Formulations

Table 2 presents a summary of the mathematical formulations and efficiency index parameters adopted in this paper. The mathematical formulations were derived from the orifice principle, which is also known as Torricelli's theorem, as well as the water balance method, the minimum night flow, fixed and variable discharge (FAVAD), burst and background estimates, unavoidable annual real losses (UARL), current annual real losses (CARL) and the infrastructure leakage index (ILI). These methodologies are supported in the literature by other studies [6,7,11,20,22,26,28,29,35–40]. Furthermore, in order to measure the effect of optimal pressure reduction and its benefits after adjusting the three PRVs to the required downstream pressures, the above hydraulic principles were used as a basis from which to derive the results for the volumetric and efficiency indexes.

**Table 2.** Summary of the methodologies for the leakage estimation indexes.

| Methodology | Mathematical Equation | Research Index Summary Advantages |
|---|---|---|
| Orifice Principle | $Q_{leak} = C h^{N1}$ <br> $Q_{leak} = C_d A \sqrt{2gh}$ | The method depends on pressure and can be applied in multiple DMAs [34,38,40]. |
| System Input Volume | $SIV = \sum\limits_{i=1}^{n} Q(l/s)_{24h} \left( \dfrac{DMA_1...DMA_6}{Average\ Flow\ Duration} \right) \times 365$ days | This index provides holistic pressure and flow data for the entire water distribution system [14,20,30]. Base data are created for developing a water balance for the DMA, supply zone or an entire bulk system [22]. |

**Table 2.** *Cont.*

| Methodology | Mathematical Equation | Research Index Summary Advantages |
|---|---|---|
| Minimum Night Flow | $L_{DMA} = Q_{MNF} - Q_{LNC}$ <br> $Q(MNF)_{DMA_1} =$ <br> $\sum_{t=1}^{n} (MNF) \left[ \begin{array}{c} (Ave.\ Night\ Flow_{0:00-4:00}) \\ - \\ (Ave.\ Nigh\ Flow_{0:00-4:00} \times DMA_{POP} \times 10l \times 6\%) \end{array} \right]$ <br> $Q_{leak(t)} = Q_L(t_{MNF}) \times \left\lvert \frac{P_{AZP(t)}}{P_{AZP}\ (t_{MNF})} \right\rvert^{\propto}$ | This is the most reliable method for estimating water leakages when consumption is at its lowest in the DMA [6,30,36]. The methodology is beneficial for assessing the effect of variable pressure on leakages during peak and off-peak periods [36]. |
| Fixed and Variable Area Discharge (FAVAD) | $Q_{leak(t)} = k_1 h^{0.5} + k_2 h^{1.5}$ <br> $Ql_i = C_i \times P_i^{\beta}$ <br> $\frac{L_1}{L_0} = \left(\frac{P_1}{P_0}\right)^{N1}$; $A = A_0 + mh$ <br> $N1 = \frac{\ln\left(\frac{Q_2^i}{Q_1^i}\right)}{\ln\left(\frac{h_2^i}{h_1^i}\right)}$ <br> $Q_{leak} = C_d\sqrt{2g}\left(A_0 h^{0.5} + mh^{1.5}\right)$ | FAVAD integrates the conservation of mass and energy, the orifice principle, the theory of hydraulics of leaks and the effect of variable pressure for leakage estimations [38]. Furthermore, it scientifically caters to turbulent flows due to pressure, material type, the type of leakage and soil hydraulics [35,41]. |
| Background and Burst Estimate (BABE) | $UARL = (18L_m + 0.8N_c + 25L_p) \times P_{AVE}$ <br> $CARL = SIV - (AC + CL)$ <br> where SIV is the system input volume (m$^3$/month); AC is the authorized consumption (m$^3$/month); and CL is the commercial loss (m$^3$/month) | BABE is beneficial in the bottom-up estimation of system leakages versus customer consumption [14,26,41]. It is a widely used method to measure CARL, ILI and UARL, producing indicative data for the FAVAD principle [6,26,36,42]. |
| Optimal Pressure Management | $\frac{BF_1}{BF_2} = \left(\frac{P_0}{P_1}\right)^{N2}$ <br> $N2 = \left(\frac{\ln\left(\frac{BF_1}{BF_2}\right)}{\ln\left(\frac{P_0}{P_1}\right)}\right)$ <br> $\Delta BF = 1 - \left(\frac{P_0}{P_1}\right)^{N2} \times (100\%) = 1 - \left(\frac{P_0 - \Delta P}{P_0}\right)^{N2} \times (100\%)$ | This index integrates the orifice and FAVAD principle through the simulation of variable pressure before and after the application of pressure management [26,43]. Pressure management is an alternative method for measuring efficiency indexes for water savings, energy savings and leakages per pipe length [40,44–46]. |
| **Efficiency Indexes** | | |
| Leakage Flow Rate | $TLD_{(n=1)} = BS_{(Date:Time)} - BF_{(Date:Time)}$ <br> where TLD is the total leakage duration (hour); BS is the basic start date and time when the service ticket was logged on SAP-PM (day or hour); and BF is the basic finish date when a leakage was physically isolated and the repair was initiated (day or hour). <br> $TAVL = (NRB \times ALFR \times ALD)$ <br> where TAVL is the total annual volume of leakage; NRB is the number of reported bursts; ALFR is the average leakage flowrate; and ALD is the average leakage duration | We used the TLD on linear repair abstracted from SAP-PM to set the benchmark for computing TLFR. Leakage durations provide base data for the estimation of real and apparent losses [18]. The method is beneficial when measuring an active leak control (ALC) component in linear leakage repair [14,36,47]. |
| Infrastructure Leakage Index (ILI) | $ILI = \frac{CARL}{UARL}$ <br> where ILI is the infrastructure leakage index; CARL is the current annual real loss (m$^3$/year); and UARL is the unavoidable annual real loss (m$^3$/year) measured as a component of SIV month by month | According to [17,20], ILI is defined as the ratio of the "current annual real losses" (CARL) to the "unavoidable annual real losses" (UARL). This dimensionless performance indicator was used in this study to assess the comprehensive leakage index in the water distribution system month by month after the reduction in optimum pressure from the PRV. |
| Total Cost of Water | Cost of water = (Volume of Water/Period) × (Water Tariff) <br> (Note that a unit cost of $3.18/m$^3$ converted from South African Rand/m$^3$ to US Dollar was used in this study) | Water is an economic resource and has a cost value [48]. Therefore, this index provides a base to estimate the cost of water production versus total losses [26]. The authors used this to estimate the total costs of water losses in the water distribution system. |
| Customer Consumption Index | $n = \frac{N}{[(1+Ne^2)]}$ <br> where $n$ is the sample size; $N$ is the total number of households; and $e$ is the level of precision at a level of 7 ± 2% | A study by [14] used this index in their study for customer meter consumption assessments. For this study, the authors sampled over 63 properties in the case study area to manually read and record water consumption levels for a period of seven days to establish consumption patterns for Phases 1 and 2. |

**Table 2.** *Cont.*

| Methodology | Mathematical Equation | Research Index Summary Advantages |
|---|---|---|
| Pressure Efficiency Index | $\Delta\%\ Pressure_{1-2} = \frac{P_1 - P_2}{P_2} \times 100$ <br><br> $\%\ Ave.\ Reduction\ Ratio_{MFF/SIV} = \left( \frac{\%\ \frac{MNF_1}{SIV_1} - \%\ \frac{MNF_2}{SIV_2}}{\%\ \frac{MNF_1}{SIV_1}} \right)$ <br><br> $IR(P_{1-2})_{(MNF/SIV} = \left( \frac{Average\ Pressure_{1-2}}{\%\ \frac{MNF_{1-2}}{SIV_{1-2}}} \right)$ | After resetting downstream operating pressures at each PRV to the required level, the team assessed the following: (1) the percentage change in pressures for Phase 1; and (2) the percentage reduction in MNF/SIV between Phases 1 and 2, as well as the index ratio (IR) of pressure versus %MNF/SIV in Phases 1 and 2. A percentage reduction in these indexes means that a change in optimal pressure has a direct positive impact on leakage control. |
| Volumetric Efficiency Index | $\%\ Reduction\ in\ MNF/SIV = m \times P_{Reduction}\ (m) + b$ <br> $\%\ Index\ ratio\ leakage = TLFR/SIV$ <br> where $m$ is the coefficient value for the linear regression; $b$ is the average constant value of MNF/SIV (l/s); $P_{Reduction}$ is the hydraulic system pressure (m); and TLFR is the total leakage flowrate volume as per the reported, unreported and leakage connections. | We used the linear regression analysis method to measure the effect of reduced pressure for the percentage reduction in MNF and SIV by volume. The assessment was carried out at each DMA and 20 critical nodal points (CNPs). Reduction by percentage ratio of MNF/SIV means that a change in optimal pressure is an alternative way to reduce the average flow during off-peak times, e.g., 12:00 a.m. and 4:00 a.m. The authors assessed the percentage index of the total leakages of TLFR/SIV before and after adjusting the PRV to optimal pressures. The reduction in the index ratio means a reduction in infrastructure leakages. |
| Index Ratio for Leakage per Kilometer | $Index\ ratio\ leakage = \sum_{n=1}^{\infty} \left( \frac{Total\ Leakages:SAP-PM}{Length\ of\ Pipeline} \right)$ | The authors further assessed the change in the sum of reported and unreported bursts per kilometer month by month for Phases 1 and 2. They used data abstracted from SAP-PM and IMQS to obtain service failures and the lengths of pipelines. A reduction in the ratio or burst pipe per kilometer indicates a reduction in AZP-reduced bursts and related leakages in water distribution systems and directly translates to water savings. |

## 3. Results and Discussion

The following sections present the analysis of the results and a discussion of the findings following the implementation of the study methodology outlined above.

### 3.1. Transient Flow Data and Pressure Analysis

In order to assess the effect of pressure on transient flow, Table 3 presents data flow analysis results for six DMAs and 20 critical nodal points (CNPs) in the water distribution system for Phases 1 and 2. The transient flow and pressure trends for the six DMAs and 20 CNPs in the distribution system are presented in Figures 3 and 4. When assessing the impact of head pressure in terms of SIV and MNF measurements, the results show that in Phase 1 of the study, the SIV was an estimated 26,272, 579 m$^3$/year, with a measured MNF of 14.01% [30], whereas Phase 2 showed a reduction in SIV to an estimated 21,915,943 m$^3$/year, with 12.50% as MNF. This reduction of bulk flow into the DMA due to changes in head pressure is equivalent to a projected 16.58% in SIV and 16.21% in MNF per year. Although nodal system output (NSO) flows look insignificant, the estimated average nodal system output (NSO) reduced from 14,774.62 m$^3$/year in Phase 1 to 12,787.85 m$^3$/year in Phase 2. The results demonstrated that the application of time-modulated PRVs in leakage control reduced the SIV and MNF percentages proportionally in LP-1, LP-2 and LP-3. The overall reduction equates to 16% of the projected annual SIV.

**Table 3.** (a) Data flow analysis results for the district metered areas (DMAs). (b) Data flow analysis results for the nodal points.

| | | | | | | (a) | | | | | | |
|---|---|---|---|---|---|---|---|---|---|---|---|---|
| | | | Phase 1 | | | | | | Phase 2 | | | |
| ID | Ave Flow (m³/s) | Ave Pressure (m) | Annual SIV (m³) | Night Flow (m³/s) | Annual MNF (m³) | Ave Flow (m³/s) | Ave Pressure (m) | Annual SIV (m³) | Night Flow (m³/s) | Annual MNF (m³) | Reduced % SIV | Reduced % MNF |
| LP-1 | 70.1 | 180 | 2,209,412 | 60.0 | 296,438 | 55.0 | 90 | 1,734,480 | 47.1 | 232,605 | 21% | 22% |
| LP-2 | 343.8 | 80 | 10,843,338 | 315.0 | 1,556,302 | 298.2 | 65 | 9,404,035 | 215.4 | 1,064,263 | 13% | 32% |
| LP-3 | 248.5 | 90 | 7,836,696 | 215.0 | 1,062,238 | 208.0 | 68 | 6,559,488 | 174.1 | 860,165 | 16% | 19% |
| LP-4.1 | 9.3 | 50 | 293,285 | 8.7 | 42,984 | 8.5 | 48 | 268,056 | 8.0 | 39,525 | 9% | 8% |
| LP-4.2 | 2.7 | 51 | 85,147 | 1.4 | 6917 | 2.7 | 45 | 83,570 | 1.38 | 6818 | 2% | 1% |
| LP-4.3 | 158.7 | 49 | 5,004,700 | 145.0 | 716,393 | 122.6 | 49 | 3,866,314 | 108.5 | 536,059 | 23% | 25% |

| | | | | | | (b) | | | | | | |
|---|---|---|---|---|---|---|---|---|---|---|---|---|
| | | | Phase 1 | | | | | | Phase 2 | | | |
| ID | Ave Flow (m³/s) | Ave Pressure (m) | Annual SIV (m³) | Night Flow (m³/s) | Annual MNF (m³) | Ave Flow (m³/s) | Ave Pressure (m) | Annual SIV (m³) | Night Flow (m³/s) | Annual MNF (m³) | Reduced % SIV | Reduced % MNF |
| 1 | 0.44 | 92.8 | 13,876 | 0.38 | 1877.4 | 0.35 | 75.0 | 11,038 | 0.27 | 1334.0 | 20% | 29% |
| 2 | 0.31 | 78.9 | 9776 | 0.31 | 1531.6 | 0.28 | 65.0 | 8830 | 0.25 | 1235.2 | 10% | 19% |
| 3 | 0.36 | 65.0 | 11,353 | 0.31 | 1531.6 | 0.33 | 63.0 | 10,407 | 0.21 | 1037.5 | 8% | 32% |
| 4 | 0.42 | 70.7 | 13,245 | 0.39 | 1926.8 | 0.41 | 60.0 | 12,930 | 0.33 | 1630.4 | 2% | 15% |
| 5 | 0.13 | 68.9 | 4100 | 0.12 | 592.9 | 0.12 | 60.0 | 3784 | 0.10 | 494.1 | 8% | 17% |
| 6 | 0.57 | 68.2 | 17,976 | 0.49 | 2420.9 | 0.56 | 55.0 | 17,660 | 0.42 | 2075.1 | 2% | 14% |
| 7 | 0.27 | 76.3 | 8515 | 0.25 | 1235.2 | 0.23 | 63.0 | 7253 | 0.18 | 889.3 | 15% | 28% |
| 8 | 0.39 | 72.0 | 12,299 | 0.34 | 1679.8 | 0.34 | 67.0 | 10,722 | 0.28 | 1383.4 | 13% | 18% |
| 9 | 0.57 | 84.3 | 17,976 | 0.49 | 2420.9 | 0.56 | 71.0 | 17,660 | 0.47 | 2322.1 | 2% | 4% |
| 10 | 0.89 | 109.2 | 28,067 | 0.66 | 3260.8 | 0.39 | 83.0 | 12,299 | 0.28 | 1383.4 | 56% | 58% |
| 11 | 0.61 | 55.9 | 19,237 | 0.51 | 2519.7 | 0.58 | 50.0 | 18,291 | 0.41 | 2025.7 | 5% | 20% |
| 12 | 0.92 | 68.2 | 29,013 | 0.66 | 3260.8 | 0.88 | 55.0 | 27,752 | 0.62 | 3063.2 | 4% | 6% |
| 13 | 0.80 | 54.5 | 25,229 | 0.71 | 3507.9 | 0.65 | 49.0 | 20,498 | 0.57 | 2816.2 | 19% | 20% |
| 14 | 0.11 | 59.4 | 3469 | 0.09 | 464.4 | 0.10 | 55.0 | 3154 | 0.08 | 395.3 | 9% | 15% |
| 15 | 0.22 | 54.3 | 6938 | 0.19 | 938.7 | 0.18 | 50.0 | 5676 | 0.15 | 741.1 | 18% | 21% |
| 16 | 0.45 | 56.7 | 14,191 | 0.31 | 1531.6 | 0.39 | 50.0 | 12,299 | 0.24 | 1185.8 | 13% | 23% |
| 17 | 0.58 | 94.0 | 18,291 | 0.50 | 2470.3 | 0.53 | 78.0 | 16,714 | 0.41 | 2025.7 | 9% | 18% |
| 18 | 0.28 | 79.7 | 8830 | 0.22 | 1086.9 | 0.26 | 73.0 | 8199 | 0.19 | 938.7 | 7% | 14% |
| 19 | 0.62 | 62.9 | 19,552 | 0.54 | 2667.9 | 0.58 | 57.0 | 18,291 | 0.50 | 2470.3 | 6% | 7% |
| 20 | 0.43 | 88.4 | 13,560 | 0.37 | 1828.0 | 0.39 | 71.0 | 12,299 | 0.32 | 1581.0 | 9% | 14% |

District metered areas were assessed and the results (presented in Figure 4) show that during Phase 1 the highest average pressure was recorded at 17.8 bar for LP-1 in contrast to 4.9 bar for LP-4.3. The two contrasting pressure results exhibited average flows of 70 l/s and 158 l/s, respectively. The latter flow was attributed to highly populated informal settlements where 15 communal standpipes were found to be leaking during visual assessments. The highest average flows were recorded as 344 l/s and 249 l/s for LP-2 and LP-3, respectively. During Phase 1, the total estimated AZP for six DMAs was 8.3 bar. After the implementation of pressure management, Phase 2 results showed a reduction in AZP, average flow and MNF.

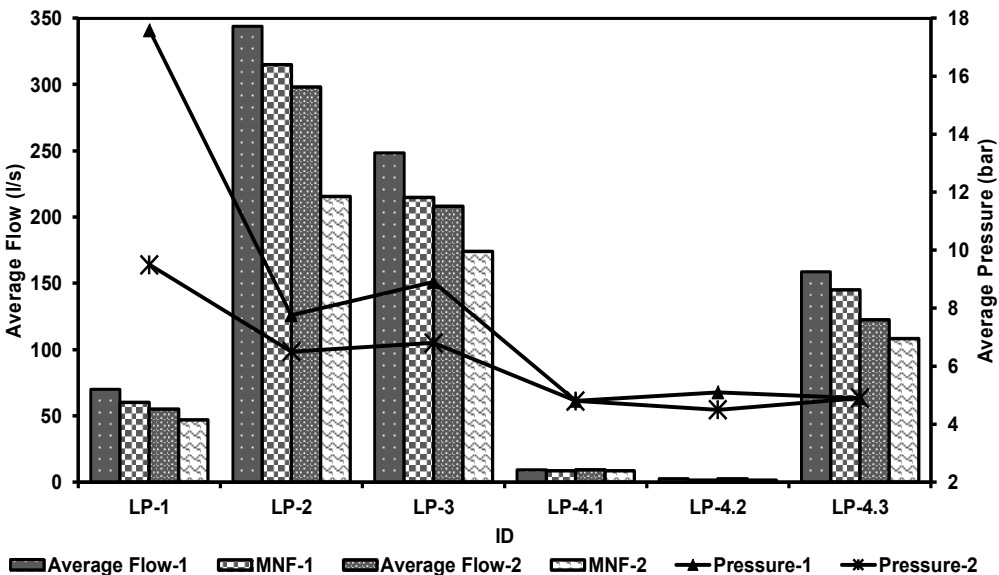

**Figure 4.** Flow and pressure for the district's metered areas.

The nodal zone points were evaluated. Figure 5 presents an analysis of the 20 critical nodal points within the six DMAs or logging points. Similar to the six DMA outcomes, higher pressure patterns were observed in all the nodal points during Phase 1 and lower pressures during Phase 2. During Phase 1, the average nodal pressure (ANP) was 7.3 bar compared to 6.3 bar during Phase 2. The results show that between Phases 1 and 2, average output flows reduced from 0.55 l/s to 0.38 per node and MNF were reduced from 0.39 l/s to 0.27 l/s. The results showed that changes in AZP at the DMA level directly influence the behavior of the nodal hydraulic flow. This finding is supported by [22,49,50].

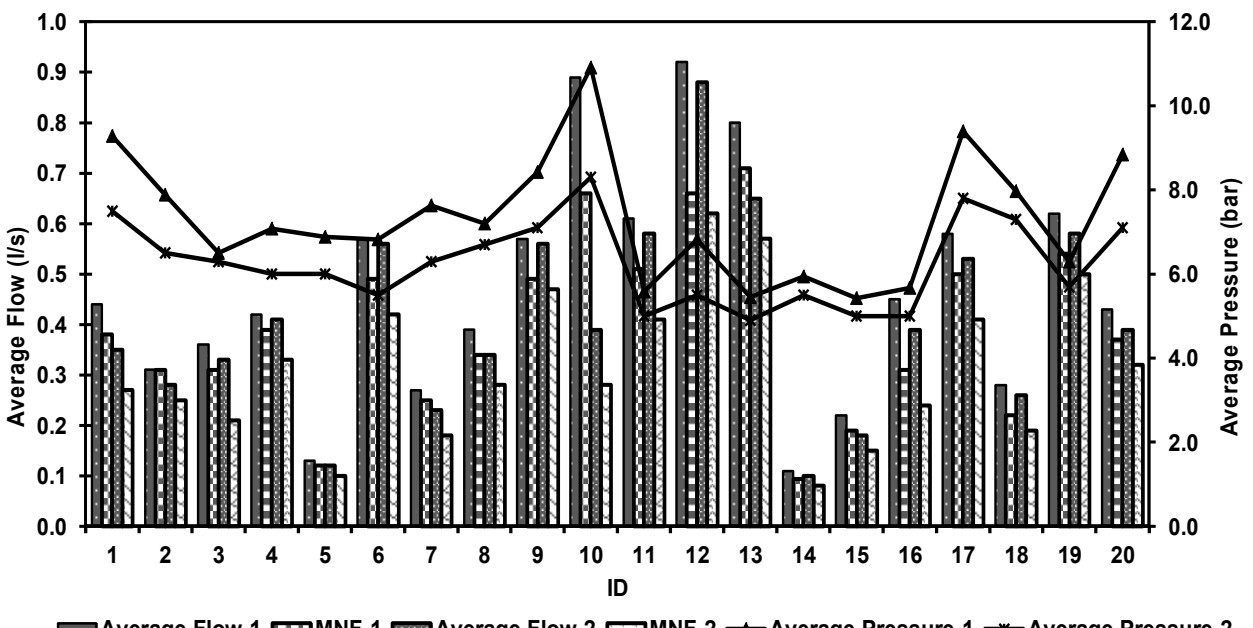

**Figure 5.** Flows and pressures for the nodes.

### 3.2. Pressure and Flow Efficiency Index Analysis

In order to measure the effects of reduced pressure, Table 4 present the efficiency indexes computed in terms of the ratio of the average pressure, MNF and SIV. This efficiency index ratio method has been applied elsewhere [49,51]. According to [49,51], the efficiency

index ratio is also called the volumetric efficiency index, where the change in pressure influences the rate of volumetric input and output. For this study, the efficiency index results are presented below. Concerning the district metered areas (DMAs), the average pressure was reduced from 83.3 m to 60.8 m. The average MNF in Phase 2 was 4.43% lower than the one for Phase 1. The average reduction percentage ratio of MFF/SIV was reduced from 13.1 to 12.4. The average index ratio (IR) representing MNF/SIV reduced from 6.5% to 5% between Phases 1 and 2. The nodal system output (NSO) was analyzed. The average pressure reduced from 73 m to 62.5 m. The average MNF in Phase 2 was 8.59% lower than that of Phase 1. The average reduction percentage ratio of MFF/SIV was reduced from 13.37 to 12.20. The average index ratio (IR) of MNF/SIV reduced from 5.51% to 5.17% between the two phases. The average ratio indexes in Phase 2 are lower than in Phase 1. Therefore, the authors preliminarily deduced that a ratio reduction in pressure has a direct efficiency index output in the percentage reduction of MNF and SIV in the water distribution system. These findings are similar to those outlined by [42,46,49,52,53] in their studies.

**Table 4.** (a) Flow and pressure efficiency index data for the district metered areas. (b) Flow and pressure efficiency index data for the nodal points.

**(a)**

| | Phase 1 | | Phase 2 | | Efficiency Index | | | |
|---|---|---|---|---|---|---|---|---|
| ID | % Ratio MNF/SIV-1 | Average Pressure-1 (m) | % Ratio MNF/SIV-2 | Average Pressure-2 (m) | % Reduction Pressure Ratio (P1–P2) | % Reduction MNF/SIV Ratio (P1–P2) | Index Ratio: Pressure-1/(%MNF/SIV-1) | Index Ratio: Pressure-2/(%MNF/SIV-2) |
| LP-1 | 13.4 | 180 | 13.4 | 90 | 50 | 0.05 | 13.4 | 6.7 |
| LP-2 | 14.4 | 80 | 11.3 | 65 | 19 | 21.15 | 5.6 | 5.7 |
| LP-3 | 13.6 | 90 | 13.1 | 68 | 24 | 3.26 | 6.6 | 5.2 |
| LP-4.1 | 14.7 | 50 | 14.7 | 48 | 4 | −0.61 | 3.4 | 3.3 |
| LP-4.2 | 8.1 | 51 | 8.2 | 45 | 12 | −0.43 | 6.3 | 5.5 |
| LP-4.3 | 14.3 | 49 | 13.9 | 49 | 0 | 3.14 | 3.4 | 3.5 |

**(b)**

| | Phase 1 | | Phase 2 | | Efficiency Index | | | |
|---|---|---|---|---|---|---|---|---|
| ID | % Ratio MNF/SIV-1 | Average Pressure-1 (m) | % Ratio MNF/SIV-2 | Average Pressure-2 (m) | % Pressure Reduction Ratio (P1–P2) | % Reduction MNF/SIV Ratio (P1–P2) | Index Ratio: Pressure-1/(%MNF/SIV-1) | Index Ratio: Pressure-2/(%MNF/SIV-2) |
| 1 | 13.53 | 93 | 12.1 | 75 | 19 | 11 | 6.86 | 6.21 |
| 2 | 15.67 | 79 | 14.0 | 65 | 18 | 11 | 5.03 | 4.65 |
| 3 | 13.49 | 65 | 10.0 | 63 | 3 | 26 | 4.82 | 6.32 |
| 4 | 14.55 | 71 | 12.6 | 60 | 15 | 13 | 4.86 | 4.76 |
| 5 | 14.46 | 69 | 13.1 | 60 | 13 | 10 | 4.76 | 4.60 |
| 6 | 13.47 | 68 | 11.8 | 55 | 19 | 13 | 5.06 | 4.68 |
| 7 | 14.51 | 76 | 12.3 | 63 | 17 | 15 | 5.26 | 5.14 |
| 8 | 13.66 | 72 | 12.9 | 67 | 7 | 6 | 5.27 | 5.19 |
| 9 | 13.47 | 84 | 13.1 | 71 | 16 | 2 | 6.26 | 5.40 |
| 10 | 11.62 | 109 | 11.2 | 83 | 24 | 3 | 9.40 | 7.38 |
| 11 | 13.10 | 56 | 11.1 | 50 | 10 | 15 | 4.26 | 4.51 |
| 12 | 11.24 | 68 | 11.0 | 55 | 19 | 2 | 6.07 | 4.98 |
| 13 | 13.90 | 54 | 13.7 | 49 | 10 | 1 | 3.92 | 3.57 |
| 14 | 13.39 | 59 | 12.5 | 55 | 7 | 6 | 4.44 | 4.39 |
| 15 | 13.53 | 54 | 13.1 | 50 | 8 | 4 | 4.01 | 3.83 |
| 16 | 10.79 | 57 | 9.6 | 50 | 12 | 11 | 5.25 | 5.19 |
| 17 | 13.51 | 94 | 12.1 | 78 | 17 | 10 | 6.96 | 6.44 |
| 18 | 12.31 | 80 | 11.4 | 73 | 8 | 7 | 6.48 | 6.38 |
| 19 | 13.65 | 63 | 13.5 | 57 | 9 | 1 | 4.61 | 4.22 |
| 20 | 13.48 | 88 | 12.9 | 71 | 20 | 5 | 6.56 | 5.52 |

### 3.3. Volumetric Linear Reduction Index

The authors analyzed the linear volumetric efficiency index ratio during Phases 1 and 2 of the study. Figure 6a,b show the change in volumetric bulk flows due to pressure assessed for the six measured DMAs and the 20 critical nodal output points. The summary outcome measurements demonstrate that the linear reduction of pressure in the distribution system resulted in a proportional reduction in the percentage ratio of MNF/SIV. When the ratio percentage of MNF/SIV is reduced due to a change in pressure, there is a higher volume of water retained in the distribution systems and losses are reduced [46,53,54]. When this happens, the water utility will likely experience reduced SIV into DMAs and nodes [30,46,47,55,56]. The respective linear identical flow ratio's constant values of 14.38

and 12.992, as well as the values of 14.298 and 12.333 for the percentage of MNF/SIV at DMAs and nodal levels, show the direct impact of changes in optimal pressure.

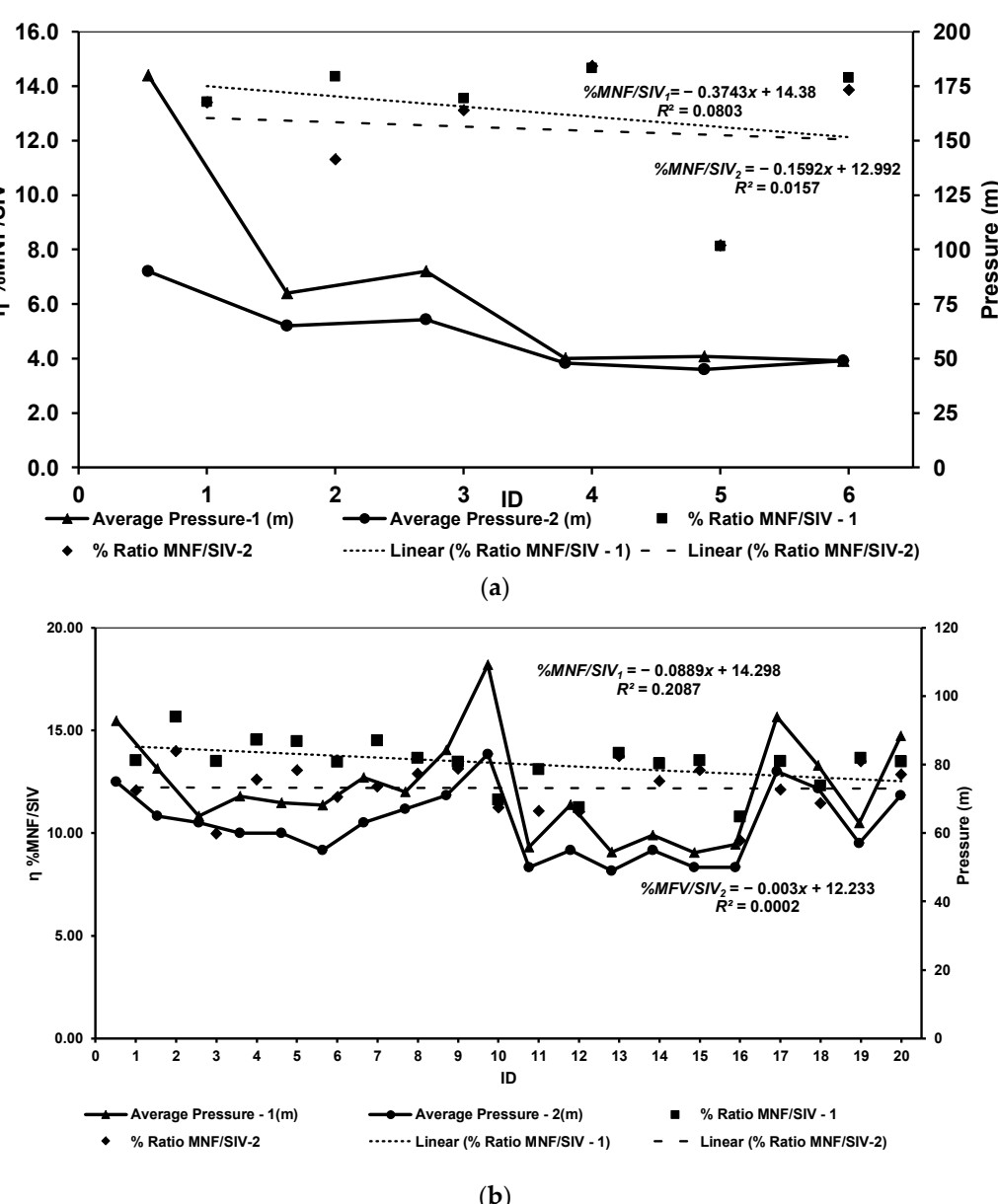

**Figure 6.** (**a**) Volumetric linear efficiency (DMAs). (**b**) Volumetric linear efficiency (nodal points).

*3.4. Leakage Flowrate Results*

3.4.1. Linear Repair Results and Indexes

To estimate the total leakage flowrates, the authors used the SAP-PM system, an operational data-centric performance information measurement software package utilized for tracking all logged service tickets from start to finish. We selected 90 job cards of reported bursts, unreported bursts and leaking connections to estimate the total leakage duration (TLD). The average leakage flowrate (ALFR) was adopted from [46]. In this study, the ALFR for the reported bursts, unreported bursts and service connections had a pressure of 240, 120 and 32 L/h/m, respectively.

Figure 7a,b present linear repairs and estimated TLFR results for reported bursts, unreported bursts and leaking service connections in the distribution system for periods between 1 June 2020 to 31 July 2021 (divided into seven months for Phase 1 and seven months for Phase 2). The leakage frequency index per kilometer of pipeline and leakage index in

relation to AZP demonstrate that all leakage ratios were lower in Phase 2 as compared to Phase 1. The results further demonstrate that, as the age of pipelines increases, reductions in pressure may reduce the leakage rate and increase the lifespan of the infrastructure. The authors preliminarily deduced that linear repair and the leakage frequency rate can be assessed through the pressure reduction comparative method, specifically when pipes are old and susceptible to handling high system pressures.

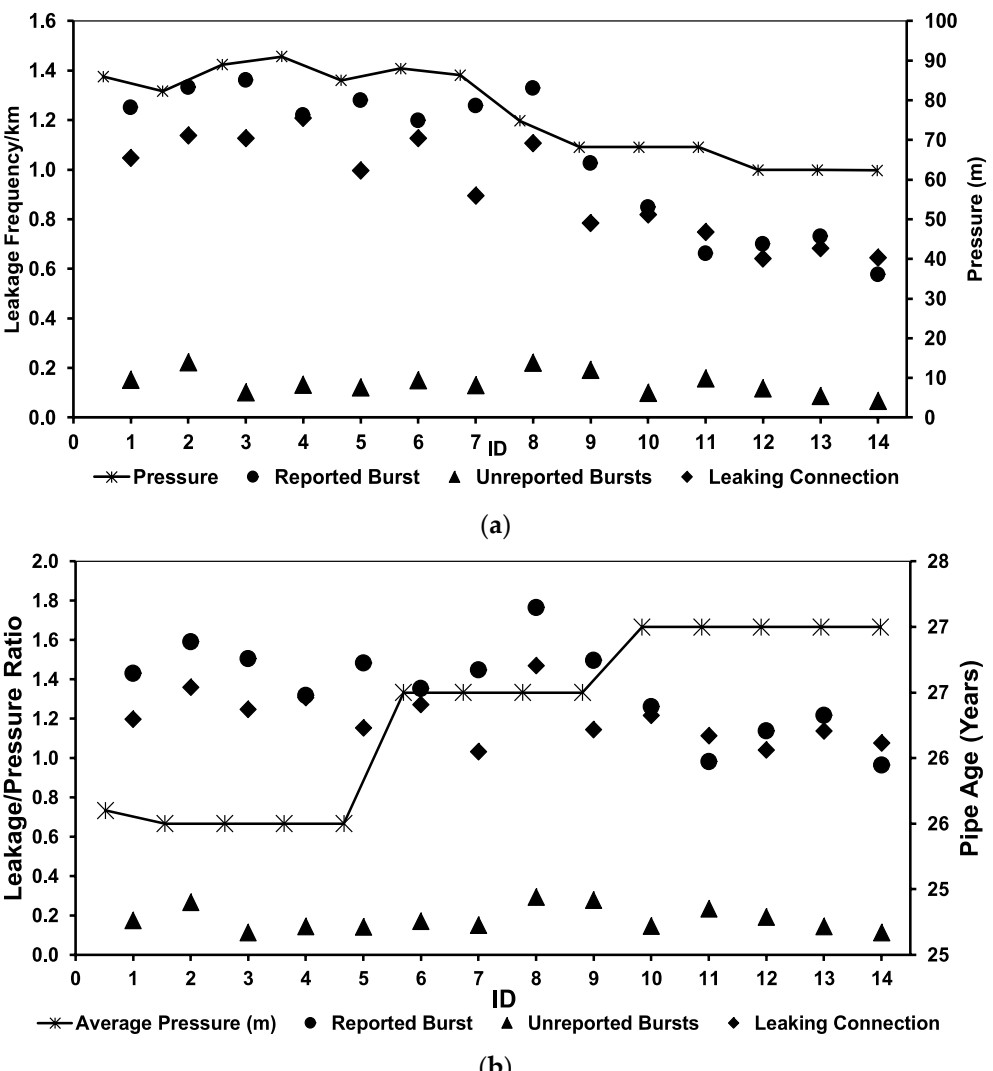

**Figure 7.** (**a**) Leakage per kilometer ratio versus average zone pressure. (**b**) Leakage and the average zone pressure ratio versus the age of the pipeline.

### 3.4.2. Leakage Estimation

Table 5 shows the linear repair data between Phase 1 and Phase 2 and the respective TLFRs. The results in Table 5 show that reported bursts (RB) and leaking connections (LC) contributed higher flowrates; however, there is a significant reduction in Phase 2 due to reduced pressures from the DMA supplying PRVs. Further results show that although unreported bursts (URBs) were fewer in number, the longer average leakage duration (ALD) makes their contribution to TLFR significant. Table 5 further shows how optimal pressure reduction reduced the average SIV from 2,189,381 m$^3$/to 1,826,329 m$^3$ between Phases 1 and 2, whereas the ratio average TLFR/SIV reduced from 0.583 to 0.497, which equates to bulk water savings of 14.71%. The bulk water loss savings are also evident in Table 6, which shows the total cost of water, as described in the proceeding section.

**Table 5.** Total leakage flow rate estimation and indexes.

| | Reported Bursts (RBs) | | | Unreported Bursts (URBs) | | | Leaking Connection (LC) | | | Linear Leakage Indexes (LLIs) | | |
|---|---|---|---|---|---|---|---|---|---|---|---|---|
| ID | RB No | ALD (hours) | TAVL (m³) | URB No | ALD (hours) | TAVL (m³) | LC No | ALD (hours) | TAVL (m³) | TLFR (m³) | SIV (m³/month) | TLFR/SIV |
| 1 | 123 | 31.20 | 921,024 | 15 | 73.55 | 132,390 | 103 | 61.01 | 201,089 | 1,254,503 | 2,189,381.50 | 0.57 |
| 2 | 131 | 31.20 | 980,928 | 22 | 73.55 | 194,172 | 112 | 61.01 | 218,660 | 1,393,760 | 2,189,381.50 | 0.64 |
| 3 | 134 | 31.20 | 1,003,392 | 10 | 73.55 | 88,260 | 111 | 61.01 | 216,708 | 1,308,360 | 2,189,381.50 | 0.60 |
| 4 | 120 | 31.20 | 898,560 | 13 | 73.55 | 114,738 | 119 | 61.01 | 232,326 | 1,245,624 | 2,189,381.50 | 0.57 |
| 5 | 126 | 31.20 | 943,488 | 12 | 73.55 | 105,912 | 98 | 61.01 | 191,327 | 1,240,727 | 2,189,381.50 | 0.57 |
| 6 | 119 | 31.20 | 891,072 | 15 | 73.55 | 132,390 | 112 | 61.01 | 218,660 | 1,242,122 | 2,189,381.50 | 0.57 |
| 7 | 125 | 31.20 | 936,000 | 13 | 73.55 | 114,738 | 89 | 61.01 | 173,756 | 1,224,494 | 2,189,381.50 | 0.56 |
| 8 | 132 | 31.20 | 988,416 | 22 | 73.55 | 194,172 | 110 | 61.01 | 214,755 | 1,397,343 | 1,826,328.58 | 0.77 |
| 9 | 102 | 31.20 | 763,776 | 19 | 73.55 | 167,694 | 78 | 61.01 | 152,281 | 1,083,751 | 1,826,328.58 | 0.59 |
| 10 | 86 | 31.20 | 643,968 | 10 | 73.55 | 88,260 | 83 | 61.01 | 162,043 | 894,271 | 1,826,328.58 | 0.49 |
| 11 | 67 | 31.20 | 501,696 | 16 | 73.55 | 141,216 | 76 | 61.01 | 148,376 | 791,288 | 1,826,328.58 | 0.43 |
| 12 | 71 | 31.20 | 531,648 | 12 | 73.55 | 105,912 | 65 | 61.01 | 126,901 | 764,461 | 1,826,328.58 | 0.42 |
| 13 | 76 | 31.20 | 569,088 | 9 | 73.55 | 79,434 | 71 | 61.01 | 138,615 | 787,137 | 1,826,328.58 | 0.43 |
| 14 | 60 | 31.20 | 449,280 | 7 | 73.55 | 61,782 | 67 | 61.01 | 130,805 | 641,867 | 1,826,328.58 | 0.35 |

### 3.4.3. Leakage Cost Indexes

Table 6 shows the total cost index parameters per leakage type as an extension of the TLFR presented in Table 5. With the year 2021's water tariff of $3.18/m³, the total combined leakage cost index between Phases 1 and 2 was $48,099,580.45, whereas the values of MNF and SIV were estimated to be $11,798,049.1 and $88,546,407.34, respectively. The results further demonstrate that the reduction in AZP between Phases 1 and 2 had a huge impact on the cost of water lost due to leakages. The results demonstrate that water is an economic good and not a social good. This assertion is supported by the authors of [21,27,48,56–60] in their studies.

**Table 6.** Leakage cost estimation index.

| | Leakage Cost Estimation | | | | % Leakage Cost Index | | | % MNF Cost Index | | % SIV Cost Index | |
|---|---|---|---|---|---|---|---|---|---|---|---|
| ID | RB | URB | LC | Total Cost | URB | LC | RB | MNF Cost | %MNF | SIV Cost | %SIV |
| 1 | $2,901,226 | $417,029 | $633,430 | $3,951,684 | 6.03% | 0.87% | 1.30% | $966,334 | 8.19% | $6,896,552 | 7.79% |
| 2 | $3,089,923 | $611,642 | $688,779 | $4,390,344 | 6.42% | 1.27% | 1.40% | $966,334 | 8.19% | $6,896,552 | 7.79% |
| 3 | $3,160,685 | $278,019 | $682,629 | $4,121,332 | 6.57% | 0.58% | 1.40% | $966,334 | 8.19% | $6,896,552 | 7.79% |
| 4 | $2,830,464 | $361,425 | $731,827 | $3,923,716 | 5.88% | 0.75% | 1.50% | $966,334 | 8.19% | $6,896,552 | 7.79% |
| 5 | $2,971,987 | $333,623 | $602,681 | $3,908,291 | 6.18% | 0.69% | 1.30% | $966,334 | 8.19% | $6,896,552 | 7.79% |
| 6 | $2,806,877 | $417,029 | $688,779 | $3,912,684 | 5.84% | 0.87% | 1.40% | $966,334 | 8.19% | $6,896,552 | 7.79% |
| 7 | $2,948,400 | $361,425 | $547,333 | $3,857,158 | 6.13% | 0.75% | 1.10% | $966,334 | 8.19% | $6,896,552 | 7.79% |
| 8 | $3,113,510 | $611,642 | $676,479 | $4,401,631 | 6.47% | 1.27% | 1.40% | $719,102 | 6.10% | $5,752,935 | 6.50% |
| 9 | $2,405,894 | $528,236 | $479,685 | $3,413,816 | 5.00% | 1.10% | 1.00% | $719,102 | 6.10% | $5,752,935 | 6.50% |
| 10 | $2,028,499 | $278,019 | $510,434 | $2,816,952 | 4.22% | 0.58% | 1.10% | $719,102 | 6.10% | $5,752,935 | 6.50% |
| 11 | $1,580,342 | $444,830 | $467,385 | $2,492,558 | 3.29% | 0.92% | 1.00% | $719,102 | 6.10% | $5,752,935 | 6.50% |
| 12 | $1,674,691 | $333,623 | $399,738 | $2,408,052 | 3.48% | 0.69% | 0.80% | $719,102 | 6.10% | $5,752,935 | 6.50% |
| 13 | $1,792,627 | $250,217 | $436,636 | $2,479,481 | 3.73% | 0.52% | 0.90% | $719,102 | 6.10% | $5,752,935 | 6.50% |
| 14 | $1,415,232 | $194,613 | $412,037 | $2,021,882 | 2.94% | 0.40% | 0.90% | $719,102 | 6.10% | $5,752,935 | 6.50% |

### 3.4.4. Customer Consumption Index

Figure 8 shows results for the consumption patterns during Phase 1 and Phase 2. We randomly selected 63 properties at each phase and manually recorded consumption for each household for a period of 7 days and used those results to estimate the average monthly consumption (AMC). The linear reduction equations for Phase 1 and Phase 2 are given as $y = 0.1601x + 31.205$ and $y = 0.1261x + 20.522$ respectively. The results show a reduction in the average consumption constant from 31.205 m³/month in Phase 1 to 20.522 m³/month in Phase 2. The results demonstrate that reduced pressure has a direct influence on customer consumption, although the authors of [11,40,53,59,60] are of the view that more studies on the influence of pressure versus consumption are needed.

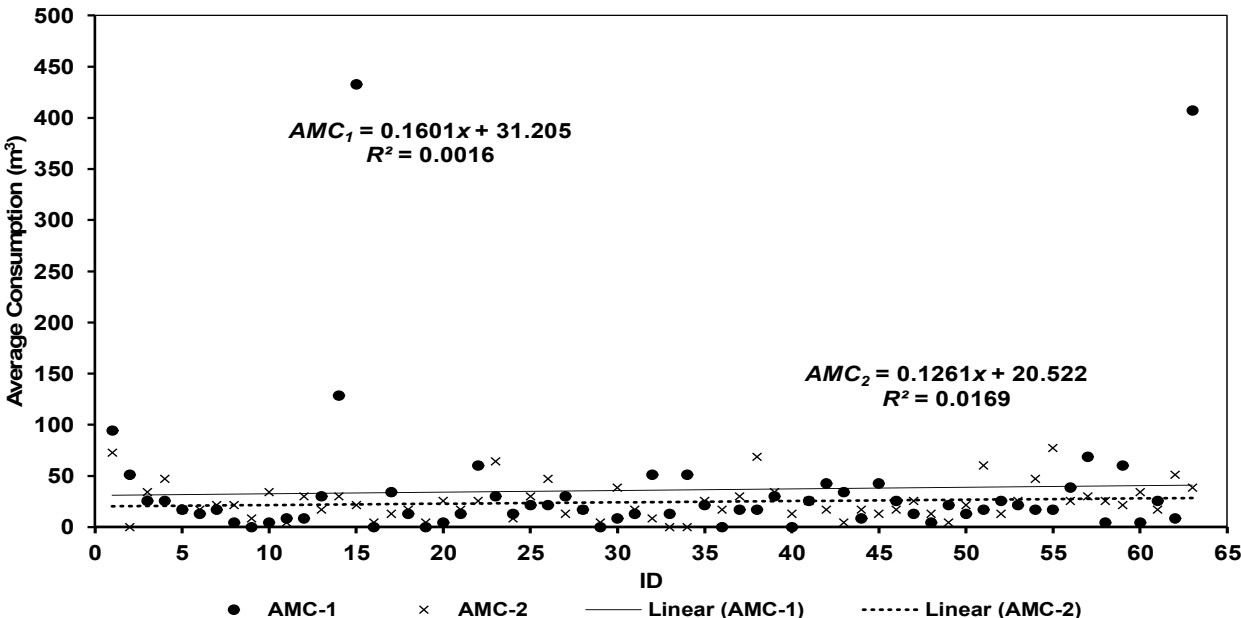

**Figure 8.** Average monthly consumption per household.

### 3.4.5. Infrastructure Leakage Index

Table 7 shows the comparative ILI between Phase 1 and Phase 2 during the study. The ILI results were established by using authorized consumption to measure the value of CARL, which is the difference between SIV and authorized consumption and commercial losses [22]. Due to the unavailability of a number of connection data from water utility, estimations for the number of connections for the computation of UARL were based on the mathematical recommendation made by the authors of [22,41,61]. The results show that ILI is almost identical between Phase 1 and Phase 2, with average ratios of 4.06 and 4.30. Although ILI in Phase 2 increased month by month compared to Phase 1, the authors concluded that ILI was influenced by SIV, consumption and reduced pressure, and therefore the outcome does not reflect an accurate finding due to many unknown factors beyond the scope of this study that need further analysis.

**Table 7.** Infrastructure leakage data indexes.

| ID | SIV | AMC | AC | CL | CARL | L (km) | N (c) | L (p) | P ($_{AVE}$) | UARL | ILI |
|----|-----|-----|----|----|------|--------|-------|-------|--------------|------|-----|
| 1 | 2,189,381 | 36.33 | 178,807 | 0 | 2,010,574 | 98,435 | 4922 | 0 | 86.0 | 490,994 | 4.1 |
| 2 | 2,189,381 | 36.33 | 178,807 | 0 | 2,010,574 | 98,435 | 4922 | 0 | 82.3 | 469,870 | 4.3 |
| 3 | 2,189,381 | 36.33 | 178,807 | 0 | 2,010,574 | 98,435 | 4922 | 0 | 89.0 | 508,122 | 4.0 |
| 4 | 2,189,381 | 36.33 | 178,807 | 0 | 2,010,574 | 98,435 | 4922 | 0 | 91.0 | 519,540 | 3.9 |
| 5 | 2,189,381 | 36.33 | 178,807 | 0 | 2,010,574 | 98,435 | 4922 | 0 | 85.0 | 485,285 | 4.1 |
| 6 | 2,189,381 | 36.33 | 180,547 | 0 | 2,008,834 | 99,393 | 4970 | 0 | 88.0 | 507,302 | 4.0 |
| 7 | 2,189,381 | 36.33 | 180,547 | 0 | 2,008,834 | 99,393 | 4970 | 0 | 86.3 | 497,502 | 4.0 |
| 8 | 1,826,329 | 36.33 | 180,547 | 0 | 1,645,781 | 99,393 | 4970 | 0 | 74.8 | 431,207 | 3.8 |
| 9 | 1,826,329 | 24.56 | 122,055 | 0 | 1,704,274 | 99,393 | 4970 | 0 | 68.2 | 393,159 | 4.3 |
| 10 | 1,826,329 | 24.56 | 124,593 | 0 | 1,701,736 | 101,460 | 5073 | 0 | 68.2 | 401,335 | 4.2 |
| 11 | 1,826,329 | 24.56 | 124,593 | 0 | 1,701,736 | 101,460 | 5073 | 0 | 68.2 | 401,335 | 4.2 |
| 12 | 1,826,329 | 24.56 | 124,593 | 0 | 1,701,736 | 101,460 | 5073 | 0 | 62.4 | 367,204 | 4.6 |
| 13 | 1,826,329 | 24.56 | 127,723 | 0 | 1,698,606 | 104,009 | 5200 | 0 | 62.4 | 376,429 | 4.5 |
| 14 | 1,826,329 | 24.56 | 127,723 | 0 | 1,698,607 | 104,009 | 5200 | 0 | 62.3 | 375,826 | 4.5 |

AMC: average monthly consumption; AC: authorized consumption; CL: commercial losses (m$^3$/month).

## 4. Conclusions and Recommendations

The purpose of this study was to demonstrate the impact of optimal pressure, its efficiency indexes for volumetric cost performance and linear leakage measurements. In a high-level setting, the study utilized FAVAD, the orifice principle, MNF and BABE methodologies in a two-phased approach in which 6 DMAs and 20 critical nodal points were used to measure hydraulic flow and pressure changes. Specifically, the results showed that changes in optimal pressure resulted in a reduction in SIV from 26,272,579 m$^3$ to 21,915,943 m$^3$, whereas MNF reduced from 14.01% to 12.50% and the average nodal system output (NSO) reduced from 14,774.62 m$^3$/year to 12,787.85 m$^3$/year. The volumetric index ratio MNF/SIV at the DMA level reduced from 13.1% to 4.3%, whereas it reduced from 13.7% to 8.9% at the NSO level. The total average leakages reduced from 246 to 177 per month, whereas leakage frequency/km/pressure reduced from 8.31% to 5.98%. The total leakage cost index reduced from \$4,009,315.54 to \$2,862,053.10. The AMC per household declined from 36.33 m$^3$ to 24.56 m$^3$, whereas the ratio of TLFR/SIV declined from 0.58 to 0.5 at a R$^2$ value of 0.4583. Finally, the month-by-month computed average ILI was 4.06 in Phase 1 and 4.30 in Phase 2. Therefore, it can be concluded that the study's findings are essential to persuade water managers, policy makers and decision makers in water utilities to invest more resources in the reduction of water losses. The assessment methods tested in this study may be used as alternative methods to measure the effect of pressure on water leakage behavior in water distribution systems due to their proven benefits in leakage control.

**Author Contributions:** Conceptualization, R.P.M. and M.S.; methodology, R.P.M.; software, R.P.M.; validation, R.P.M., M.S. and S.N.-B.; formal analysis, R.P.M.; investigation, R.P.M.; resources, R.P.M.; data curation, R.P.M.; writing—original draft preparation, R.P.M.; writing—review and editing, M.S. and S.N.-B.; visualization, R.P.M.; supervision, M.S. and S.N.-B.; project administration, R.P.M.; funding acquisition, R.P.M. and M.S. All authors have read and agreed to the published version of the manuscript.

**Funding:** This research was supported by RainSolutions (Water JPI 2018 Joint Call project) via collaboration between Johannesburg University and Lund University.

**Institutional Review Board Statement:** The study was conducted according to the guidelines of the Declaration of the Department of Higher Education and Training of South Africa Government Gazette, No. 39583, Vol 607 of 08 January 2016 and originally approved by the University of Johannesburg on 12 February 2018 and the Institutional Review and Ethical committee of the University of Johannesburg on 24 August 2020 with Ethical Clearance Number: UJ_FEBE_FEPC_00034.

**Informed Consent Statement:** Informed consent was obtained from all subjects involved in the study and approved under Ethical Clearance Number: UJ_FEBE_FEPC_00034.

**Data Availability Statement:** All generated and collected data, models, or code used during the study were provisionally and ethically granted by Johannesburg Water SOC Ltd. and the University of Johannesburg. Some or all data, models, or code that support the findings of this study are available from the corresponding authors upon reasonable request.

**Acknowledgments:** The authors acknowledge the academic support from the University of Johannesburg and the data provided by Johannesburg Water SOC Ltd. (Johannesburg Metropolitan Municipality). This work was also supported by RainSolutions (Water JPI 2018 Joint Call project).

**Conflicts of Interest:** The authors declare no conflict of interest.

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
