# Peer review of "Optimal Pressure Management in Water Distribution Systems: Efficiency Indexes for Volumetric Cost Performance, Consumption and Linear Leakage Measurements"

_water, doi:10.3390/w14050805_

Round 1
Reviewer 1 Report
This paper presents a very interest study on optimal pressure management of Water Distribution Networks (WDN) with a particular focus on developing countries. Specifically, the authors compare and assess different management strategies in a real case. This and a data collection campaign make the paper particularly interesting. Despite I consider this topic of paramount importance, I think that this work have still some important flaws especially related to readability and reproducibility of the work. I have no doubt that the authors will be able to re-design properly the work in order to make the paper publishable on WATER.
Here some suggestion:
The methodology adopted is not properly presented and in general the paper is difficult to read. Does the author use an hydraulic model? Is it base on EPANET 2.2 or 2? Does the author collect the data and then calibrate the model? Does the author compare the model with the data collected? Are the pressure management strategies implemented in the real case and used to test the model?
I would suggest to write properly the methodological section. First, all the tools, model adopted, and the application. Application and methods are indeed mixed and this make the paper difficult to read.
There are a lot of acronym used and not properly introduced.
Line 43: infrastructural losses and leakages [4,11] please add the following reference: “Burst Detection in Water Distribution Systems: The Issue of Dataset Collection, Applied Sciences”
Line 94: The authors refer to “use of hydraulic modelling software applications recommended by [33]”. Could the authors be more specific? Could the author dedicate a specific paragraph on this? Does the author use “WADISO”? Is this software base on EPANET 2 or 2.2? I would like to see section dedicated to the hydraulic simulation as well as a section dedicated to the input data.
Figure 1: Please provide a more understandable figure. I would suggest to use a block diagram
Figure 2: a 2D maps could result more readable
Line 50: please add the following reference “Global Gradient Algorithm extension to distributed pressure driven pipe demand model, 2019”
Line 51: please introduce each acronym use in the paper
Please provide a figure of the visual condition assessment
Please add a table with the WDN characteristics: pipes, nodes, PRV, elevations ecc…
Author Response
Response to Reviewer 1 Comments
Review Report Form
Open Review
(x) I would not like to sign my review report
( ) I would like to sign my review report
English language and style
( ) Extensive editing of English language and style required
( ) Moderate English changes required
(x) English language and style are fine/minor spell check required
( ) I don't feel qualified to judge about the English language and style
|
Yes |
Can be improved |
Must be improved |
Not applicable |
|
|
Does the introduction provide sufficient background and include all relevant references? |
(x) |
( ) |
( ) |
( ) |
|
Is the research design appropriate? |
( ) |
( ) |
(x) |
( ) |
|
Are the methods adequately described? |
( ) |
( ) |
(x) |
( ) |
|
Are the results clearly presented? |
(x) |
( ) |
( ) |
( ) |
|
Are the conclusions supported by the results? |
(x) |
( ) |
( ) |
( ) |
Comments and Suggestions for Authors
Is the research design appropriate? Must be improved
REPLY:
- The research design is appropriate and has been improved, re-aligned and edited.
- Reference to this is page 6-9 from line 291-891.
Are the methods adequately described? Must be improved
REPLY: The has been adequately described and has been improved. Reference to this is in page 3,4,5,6 and 7.
This paper presents a very interest study on optimal pressure management of Water Distribution Networks (WDN) with a particular focus on developing countries. Specifically, the authors compare and assess different management strategies in a real case. This and a data collection campaign make the paper particularly interesting.
Despite I consider this topic of paramount importance, I think that this work have still some important flaws especially related to readability and reproducibility of the work. I have no doubt that the authors will be able to re-design properly the work in order to make the paper publishable on WATER.
REPLY:
- The authors have re-designed the paper for readability and producibility.
- The entire manuscript was revised, repackaged and edited from line 23 page 1 to line 1512 in page 23. Figure 1 (page 4) and Figure 2 (page 5) were reproduced.
- Figure 3 (Page 6) and the corresponding write-up from page 6 and 7 was added to increase the quality of the manuscript.
Here some suggestion:
The methodology adopted is not properly presented and in general the paper is difficult to read.
REPLY:
- The methodology had been improved and is properly presented (page 6 to page 9)
Does the author use an hydraulic model? Is it base on EPANET 2.2 or 2?
REPLY:
- The hydraulic mode used is WADISO, a group of EPANET 2.2. This is described in line 302- 325 (page5)
Does the author collect the data and then calibrate the model? Does the author compare the model with the data collected? Are the pressure management strategies implemented in the real case and used to test the model?
REPLY:
- Yes, Authors collect data and compare the findings with the existing model. This is addressed in page 3, 4, 5 an 6.
- The results for phase 1(before) and phase 2(after) are presented in page 11, 12, 13 and 14 are a proof of the real case study findings of the tested model
I would suggest to write properly the methodological section. First, all the tools, model adopted, and the application. Application and methods are indeed mixed and this make the paper difficult to read.
REPLY:
- The methodology has been revised, referenced and chronologically with reference to pre-data collection, method development and mathematical formulation (page 6, 7,8 and 9).
- All tools and model adopted addressed (page 3,4 and 5)
There are a lot of acronym used and not properly introduced.
REPLY:
- All acronyms used were introduced first before further usage in the manuscript; e.g., total leakage flowrate (TLFR)- page 1, critical nodal point (CNP) page 748.
Line 43: infrastructural losses and leakages [4,11] please add the following reference: “Burst Detection in Water Distribution Systems: The Issue of Dataset Collection, Applied Sciences”
REPLY: Reference added as [3] and appeared in the reference list page 20
Line 94: The authors refer to “use of hydraulic modelling software applications recommended by [33]”. Could the authors be more specific? Could the author dedicate a specific paragraph on this? Does the author use “WADISO”? Is this software base on EPANET 2 or 2.2?.
REPLY:
- The text has been revised and more information on the methodology has been provided. The hydraulic modelling in question has been answered. WADISO is a hydraulic modelling application and is within the group of EPANENT 2.2. This information has been added and aligned (Line 289 to 301) corresponding page. (4)
I would like to see section dedicated to the hydraulic simulation as well as a section dedicated to the input data
REPLY:
- This has been addressed in section 2.2 (Pages 3,4 and 5)
Figure 1: Please provide a more understandable figure. I would suggest to use a block diagram
REPLY:
- Figure 1 has been changed and presented in more simpler block structure (page 4).
Figure 2: a 2D maps could result more readable
REPLY:
- Figure 2 has already been presented in 2D in the original manuscript. We have now made it more readable and added a legend. The figure format has also been changed (page 5 ).
Line 50: please add the following reference “Global Gradient Algorithm extension to distributed pressure driven pipe demand model, 2019”
REPLY:
- This reference has been added as [12] on Line 54, 58 and 69
- Reference list (page 21).
Line 51: please introduce each acronym use in the paper
REPLY:
- All acronyms used were introduced first before further usage in the manuscript; e.g., total leakage flowrate (TLFR)- page 1, average zonal pressure (AZP) page 3, critical nodal point (CNP) page 748.
Please provide a figure of the visual condition assessment
REPLY:
- Figure 3 has been added and further supported by Table 1 (page 6, 7).
Please add a table with the WDN characteristics: pipes, nodes, PRV, elevations ecc…
REPLY:
- Figure 3 has been added and further supported by Table 1 (a, b, c) (Page 7).

Reviewer 2 Report
Minor comments
- It would be nice to have the paper proof-read by a native speaker, I can see some sun on sentences throughout the paper, such as in lines 2-7 of the abstract.
- Mechanics of writing should be attended to a little more closely before the next revision, such as full stops, commas, brackets and so on throughout the paper.
- What is TLFR in the abstract?
- MNF is not defined in the abstract before the first time it is used.
- Please capitalize the acronyms when defining. Also, when using acronyms there is no need to repeat the definition. Please correct throughout the paper.
- What is AZP on line 51?
- There are no legends in figure 2. What are the dots and colored lines?
Major Comments
I’m not convinced that this is a novel idea able to be published at this point because of the following:
- The authors are using PRVs for pressure management. This is a good approach if the system is over pressured. Which causes huge energy waste in the system. while I acknowledge that in some cases because of the topography PRVs is the only solution available, I suggest using merely PRVs, while ignoring the operational pressure regime in the system, generally, is not preferred as a n energy stand point. I believe this pressure management approach should also be looked at, considering pressure management at the pump station, in order to put in less pressure to the system in the first place. Which is more about operation. With that I would suggest adding more discussions in the introduction about this and adding some more papers in the literature that are related to the leakage and pressure management on the pump station side such as below:
- Dini, M., Hemmati, M., Hashemi, S. (2022). “Optimal Pump Scheduling to Improve Network Reliability and Leakage in Water Distribution Networks”, Springer Journal of Water Resources Management.
- Hashemi, S. S., Tabesh, M., & Ataeekia, B. (2013). Scheduling and operating costs in water distribution networks. Proceedings of the ICE-Water Management, 166(8), 432-442.
- Introduction is way too short and does not cover energy that is closely related to leakage and pressure management like I said above. It is not clear what the novelty of this paper is, as using PRVs in terms of controlling pressure and leakage in a real-world system while is not quite an academic or novel idea these days.
- Also, when it comes to the idea of pressure management to reduce pipe burst there is more to be said in the literature review such as the following papers that are missing:
- Shirzad, A., Tabesh, M., & Farmani, R. (2014). A comparison between performance of support vector regression and artificial neural network in prediction of pipe burst rate in water distribution networks. KSCE Journal of Civil Engineering, 18(4), 941-948.
- Shirzad, A., & Safari, M. J. S. (2019). Pipe failure rate prediction in water distribution networks using multivariate adaptive regression splines and random forest techniques. Urban Water Journal, 16(9), 653-661.
Author Response
Response to Reviewer 2
REVIEWER 2
Review Report Form
Open Review
(x) I would not like to sign my review report
( ) I would like to sign my review report
English language and style
( ) Extensive editing of English language and style required
(x) Moderate English changes required
REPLY: Moderate English changes has been implements over and above the MDPI language editorial work throughout the manuscript.
( ) English language and style are fine/minor spell check required
( ) I don't feel qualified to judge about the English language and style
|
Yes |
Can be improved |
Must be improved |
Not applicable |
|
|
Does the introduction provide sufficient background and include all relevant references? |
( ) |
( ) |
(x) |
( ) |
|
Is the research design appropriate? |
( ) |
( ) |
(x) |
( ) |
|
Are the methods adequately described? |
( ) |
( ) |
(x) |
( ) |
|
Are the results clearly presented? |
( ) |
(x) |
( ) |
( ) |
|
Are the conclusions supported by the results? |
( ) |
(x) |
( ) |
( ) |
Does the introduction provide sufficient background and include all relevant references? Must be improved
REPLY:
- The introduction provide sufficient background and includes reference. It has also been improved. Reference to line 68-80 (page 2)
Is the research design appropriate? Must be
REPLY:
- The research design is appropriate and has been improved, re-aligned and edited.
- Reference to this is page 6-9 from line 291-891.
Are the methods adequately described? Must be improved
REPLY:
- The has been adequately described and has been improved. Reference to this is in page 3,4,5,6 and 7.
Are the results clearly presented? Can be improved
REPLY:
- The results are clearly presented and improved in page 11 and all references aligned throughout the manuscript. All the results were compared against other research work in some parts of the world. Page 11,12, 13, 14, 15, 16, 17 and 18
Are the conclusions supported by the results? Can be improved
REPLY:
- The conclusion is supported by the results and has been improved reference to ** line 1342-1345, (on page 19)
Comments and Suggestions for Authors
Minor comments
- It would be nice to have the paper proof-read by a native speaker, I can see some sun on sentences throughout the paper, such as in lines 2-7 of the abstract.
REPLY:
- The manuscript was first proof read and edited by MDPI language experts and was issued with a certificate of compliance on language usage. Moreover, a native English speaker has now edited the revised article.
- Mechanics of writing should be attended to a little more closely before the next revision, such as full stops, commas, brackets and so on throughout the paper.
REPLY:
- The manuscript was first proof read and edited by MDPI language experts and was issued with a certificate of compliance on language usage. The usage of comas and brackets was further reviewed by the authors.. Moreover, a native English speaker has now edited the revised article.
- What is TLFR in the abstract?
REPLY:
- The acronym TLFR stands for total leakage flowrate (page 1).
- The text has been modified in the revised manuscript and all acronyms were introduced the first time that they were used.
- All acronyms used were introduced first before further usage in the manuscript; e.g., total leakage flowrate (TLFR)- page 1, critical nodal point (CNP) line 748.
- MNF is not defined in the abstract before the first time it is used.
REPLY:
- MNF is the acronym for Minimum Night Flow. The text has been modified in the revised manuscript and all acronyms were introduced the first time that they were used (Page 1, 4, 6, 8, 9 to 20).
- Please capitalize the acronyms when defining. Also, when using acronyms there is no need to repeat the definition. Please correct throughout the paper.
REPLY:
- The manuscript was first proof read and edited by MDPI language experts and was issued with a certificate of compliance on language usage. The decision to not capitalize acronyms in the text was made by our MDPI reviewers. Furthermore, definition repetitions have been avoided.
- What is AZP on line 51
REPLY:
- AZP has been defined as average zonal pressure (Line 55, page 2 ).
- There are no legends in figure 2. What are the dots and colored lines?
REPLY:
- The legend for Figure 2 has been added. The dots represent logging points where ultra-sonic flow logging devices were positioned. These dots represents DMA and nodes in the distribution system. The color lines are meant to make it easier for the reader to see how each DMA was zoned and demarcated during the study. Figure 2 is presented in 2D for easy readability (page 5).
Major Comments
I’m not convinced that this is a novel idea able to be published at this point because of the following:
- The authors are using PRVs for pressure management. This is a good approach if the system is over pressured. Which causes huge energy waste in the system. while I acknowledge that in some cases because of the topography PRVs is the only solution available, I suggest using merely PRVs, while ignoring the operational pressure regime in the system, generally, is not preferred as a n energy stand point. I believe this pressure management approach should also be looked at, considering pressure management at the pump station, in order to put in less pressure to the system in the first place. Which is more about operation. With that I would suggest adding more discussions in the introduction about this and adding some more papers in the literature that are related to the leakage and pressure management on the pump station side such as below:
REPLY:
- The manuscript discusses a representative case study to address serious socio-economic issues in a developing country. The study is highly relevant as it introduces the non-traditional flow-modulated PRV, which were in operation by introducing a time-modulated PRV that gave the water utility more privilege to reduce water losses between 12:00 AM and 4:00 AM. The area was high pressured zone due to the topographical layout (Reference table 1 a, b and c). There is also a high withdrawal of water due to prevailing socio-economic and unattended domestic background leakages in the area. The study also addressed the effects of pressure reduction on domestic consumption in the study area, which highlights the misconception that all water losses after metering devices were a component of apparent losses. The study opens new avenues for reducing domestic background losses in future studies. We also covered total cost methods and the effects of water losses in this paper, which opens further avenues for future cost and benefit analyses (page 7, 11, 18, 19).
- Dini, M., Hemmati, M., Hashemi, S. (2022). “Optimal Pump Scheduling to Improve Network Reliability and Leakage in Water Distribution Networks”, Springer Journal of Water Resources Management.
- Hashemi, S. S., Tabesh, M., & Ataeekia, B. (2013). Scheduling and operating costs in water distribution networks. Proceedings of the ICE-Water Management, 166(8), 432-442.
REPLY:
- The two sources below are now referenced in the manuscript to encourage further studies as reference [16] and [25] (page 2 and 18)
- Introduction is way too short and does not cover energy that is closely related to leakage and pressure management like I said above. It is not clear what the novelty of this paper is, as using PRVs in terms of controlling pressure and leakage in a real-world system while is not quite an academic or novel idea these days.
REPLY:
- The energy component was not the study focus of this paper, although it is interrelated. We are of the view that energy consumption as a component of water production, transmission and pricing should be the focus areas for further exploration. Moreover, we indicated in our literature how other studies are looking into these components and the revised manuscript covers this (page 2).
- Also, when it comes to the idea of pressure management to reduce pipe burst there is more to be said in the literature review such as the following papers that are missing:
- Shirzad, A., Tabesh, M., & Farmani, R. (2014). A comparison between performance of support vector regression and artificial neural network in prediction of pipe burst rate in water distribution networks. KSCE Journal of Civil Engineering, 18(4), 941-948.
- Shirzad, A., & Safari, M. J. S. (2019). Pipe failure rate prediction in water distribution networks using multivariate adaptive regression splines and random forest techniques. Urban Water Journal, 16(9), 653-661.
REPLY:
- The sources below have been referenced in the revised article (page 2 and 9).

Reviewer 3 Report
Dear Authors, the Figure 1. Process flow and analysis methods for the case study' is not readable in pdf format - can you present it in original format ? Possible explanation for the numbers in Table 2 of "Reduced % SIV and Reduced % MNF" would be interesting to the scientific audience and practical use in the future. Conclusions and recommendations section statement " The tested assessment methods should be used as an alternative to measuring the effect of pressure on water leakage behavior." can be explained with main advantages of related methods. Sincerely, Reviewer.
Author Response
RESPONSE TO REVEIWER -3
Review Report Form
Open Review
(x) I would not like to sign my review report
( ) I would like to sign my review report
English language and style
( ) Extensive editing of English language and style required
( ) Moderate English changes required
( ) English language and style are fine/minor spell check required
(x) I don't feel qualified to judge about the English language and style
|
Yes |
Can be improved |
Must be improved |
Not applicable |
|
|
Does the introduction provide sufficient background and include all relevant references? |
(x) |
( ) |
( ) |
( ) |
|
Is the research design appropriate? |
(x) |
( ) |
( ) |
( ) |
|
Are the methods adequately described? |
( ) |
(x) |
( ) |
( ) |
|
Are the results clearly presented? |
(x) |
( ) |
( ) |
( ) |
|
Are the conclusions supported by the results? |
(x) |
( ) |
( ) |
( ) |
Are the methods adequately described? Can be improved
REPLY:
- The has been adequately described and has been improved. Reference to this is in page 3,4,5,6 and 7.
Comments and
Suggestions for Authors:
Dear Authors, the Figure 1. Process flow and analysis methods for the case study' is not readable in pdf format - can you present it in original format ?
REPLY:
- Figure 1 has improved (4).
Possible explanation for the numbers in Table 2 of "Reduced % SIV and Reduced % MNF" would be interesting to the scientific audience and practical use in the future.
REPLY:
- The terms Reduced %SIV and Reduced %MNF due to reduction in pressure have been explained in the revised manuscript ** Line 1122-1126 (page 10,11)
Conclusions and recommendations section statement " The tested assessment methods should be used as an alternative to measuring the effect of pressure on water leakage behavior." can be explained with main advantages of related methods. Sincerely, Reviewer.
REPLY:
- The quoted statement has been revised **Line 1344-1346 (page 19).
Round 2
Reviewer 1 Report
The authors answered properly to all my requests
Reviewer 2 Report
It seems that the authors have covered my points based on their responses. I have no issues with the paper being published if the editor is fine with the novelty.